# STIM-Orai1 signaling regulates fluidity of cytoplasm during membrane blebbing

Kana Aoki [1], Shota Harada[2], Keita Kawaji[2], Kenji Matsuzawa [1], Seiichi Uchida [2] & Junichi Ikenouchi [1✉]

The cytoplasm in mammalian cells is considered homogeneous. In this study, we report that the cytoplasmic fluidity is regulated in the blebbing cells; the cytoplasm of rapidly expanding membrane blebs is more disordered than the cytoplasm of retracting blebs. The increase of cytoplasmic fluidity in the expanding bleb is caused by a sharp rise in the calcium concentration. The STIM-Orai1 pathway regulates this rapid and restricted increase of calcium in the expanding blebs. Conversely, activated ERM protein binds to Orai1 to inhibit the store-operated calcium entry in retracting blebs, which results in decreased in cytoplasmic calcium, rapid reassembly of the actin cortex.

[1] Department of Biology, Faculty of Sciences, Kyushu University, Fukuoka 819-0395, Japan. [2] Department of Advanced Information Technology, Kyushu University, Fukuoka 819-0395, Japan. ✉email: ikenouchi.junichi.033@m.kyushu-u.ac.jp

Amoeboid cell movement is a universal mode of cell migration found throughout the animal kingdom[1,2]. Not only the protozoan ameba but also the primordial germ cells and immune cells of higher vertebrates migrate using this mode of motion[3]. Recently, it was shown that invasive cancer cells move through tissues by amoeboid cell movement[4,5]. Amoeba-like cell movement is driven by the formation of cell membrane blebs. Membrane blebs are spherical protrusions of the plasma membrane that detach from the underlying actin filaments[6]. The mechanisms involved in the formation and retraction of membrane blebs remains largely unknown[7,8]. Whereas actin polymerization underneath the plasma membrane causes the plasma membrane to protrude in lamellipodia, the plasma membrane protrudes by the rapid influx of the cytoplasm to the actin cortex-free membrane region in blebs. The influx of cytoplasm is caused by the difference between the pressure inside the cell and the pressure outside the cell[9]. Therefore, it is necessary to compartmentalize the cytoplasm into regions of high fluidity and low fluidity in order to restrict the region where intracellular pressure is relieved and to produce cytoplasmic flow towards the protrusion at once. In fact, in protozoan amoeba, it is observed that the cytosol flows in the direction of pseudopod extension[10,11]. However, there is an ongoing debate about how cytoplasmic fluidity is locally regulated[12,13]. In this study, we elucidated the mechanism of heterogenization of cytoplasm focusing on amoeboid cell movement of cancer cells.

## Results

**Highly fluid cytoplasm is formed in the expanding membrane blebs.** We first compared the velocity of membrane deformation during membrane blebbing of human colorectal carcinoma cell line DLD1 cells. The speed of expansion of membrane blebs is much faster than that of retraction of membrane blebs (Fig. 1a, b). Then we examined whether the fluidity of cytoplasm was altered in the expanding blebs. In order to visualize the fluidity of cytoplasm, fluorescent quantum dots (QDs) were introduced into DLD1 cells by electroporation to perform particle tracking analysis in the cytoplasm (Fig. 1c and Supplementary Movie 1). First, we examined whether the motility of QDs introduced into the cytoplasm is regulated by actin networks. Destabilization of actin network in the cytoplasm by treating cells with Latrunculin B increased QDs mobility as previously reported[14] (Fig. 1d–f). Then, we compared the mobility of QDs in the cytoplasm of expanding blebs and retracting blebs. The effects of convective flows derived from the influx of cytoplasm into the expanding blebs on the mobility of QDs was minimized by comparing the mobility of QDs at the late stage of expanding blebs and at the beginning of retracting blebs. The late stage of expansion phase is defined here as the period after the maximum expansion of the bleb and before the start of reassembly of actin cortex; the period after reassembly of actin cortex and before the bleb regresses to 80% of the maximum expansion is defined as the early stage of retraction. Intriguingly, the mobility of QDs decreased sharply in retracting blebs (Fig. 1c, d). The spatial extent of QDs mobility by random diffusion was measured by the mean square displacement (MSD; Fig. 1e), which showed that the diffusion coefficient (D) of QDs was greater in the expanding blebs than in the retracting blebs (Fig. 1f). Based on these results, we concluded that cytoplasmic fluidity is increased in the expanding blebs.

Next, we examined whether the cytoplasmic region of expanding blebs is qualitatively different from non-bleb cytoplasm in terms of protein composition. Using a library of expression vectors focused on genes involved in the regulation of actin filaments, we searched for proteins that accumulate in the bleb cytoplasm only during the expansion phase. As a result, we found that Mena, VASP and PIP5K gamma accumulate only in the cytoplasm during the expansion phase (Fig. 1g–k, Supplementary Fig. 1a–f, and Supplementary Movie 2). At present, its physiological significance and molecular mechanism are unclear. However, one possibility is that rapid and extensive fragmentation of actin filaments during expansion phase creates a temporary surplus pool of actin filament plus ends, causing Mena, normally present at the actin cortex, to appear to accumulate in the cytoplasm. Thus, a highly fluid cytoplasm containing a distinctive set of proteins exists in expanding blebs.

**The concentration of calcium ions increases in the expanding membrane blebs.** Next, we investigated whether the concentration of cytoplasmic calcium ions oscillate in association with the drastic changes of cytoplasmic fluidity during membrane blebbing. GCaMP6s was expressed in DLD1 cells to visualize the change in the concentration distribution of calcium ions. We found that the concentration of calcium ions in the cytoplasm of the expanding bleb is higher than in other cytoplasmic regions (Fig. 2a–d and Supplementary Movie 3). We also confirmed that the concentration of calcium ions in the vicinity of the plasma membrane is higher in the expanding bleb than in the retracting bleb by using GCaMP6s localized to the plasma membrane (Fig. 2e, f). The concentration of calcium ions in the cytoplasm was elevated during the expansion phase, but decreased during the retraction phase. It was reported that myosin regulatory light chain (MRLC) is recruited to the actin cortex during the retraction phase significantly later than other actin associated proteins[8]. The calcium ions are elevated only in the cytoplasm of expanding blebs which lacks acto-myosin cortex, and the calcium ion concentration at the time of myosin accumulation in the retracting bleb is as low as that in the cell body (Supplementary Fig. 2). The calcium ion concentration in the cell body was constant throughout the time course of the bleb expansion and retraction (Fig. 2b, c). Therefore, we reasoned that the increase in the concentration of calcium ions in the cytoplasm of the expanding bleb is not related to the regulation of myosin contractility during the bleb cycle but to the increase of the cytoplasmic fluidity in the expanding bleb.

Given these observations, we treated the cells with a non-fluorescent calcium ionophore, 4-bromo-A23187, to examine whether the concentration of calcium ions in the cytoplasm controls the bleb cycle. The artificial influx of calcium ions into the cytoplasm by treatment with 4-bromo-A23187, prolonged the bleb expansion period, increased bleb size, and induced all blebs across the plasma membrane to eventually fuse (Fig. 3a, f–h, Supplementary Fig. 3a, b, and Supplementary Movie 4). Moreover, the fluidity of the overall cytoplasm increased to the same extent as the cytoplasm of the expanding bleb by 4-bromo-A23187 treatment (Fig. 3b–e). Similarly, inhibition of cytoplasm-to-ER calcium ion uptake by treatment with an inhibitor of the calcium ion transporter SERCA, thapsigargin, increased the cytoplasmic concentration of calcium ions and induced the formation of enlarged blebs (Fig. 3i). These findings indicate that restricted up-regulation of calcium concentration in cytoplasm triggers increase of cytoplasmic fluidity required for bleb expansion.

**The influx of calcium ions via SOCE support expansion of membrane blebs.** What is the molecular mechanism responsible for the calcium ion elevation in expanding blebs? We found that removal of calcium ion in the medium immediately decreases the bleb size and shortens the expansion phase of the bleb cycle (Supplementary Fig. 3c–h). Therefore, we speculated that the influx of calcium from outside the cell plays an important role in bleb expansion. Store-operated calcium entry (SOCE) is a major mechanism of calcium ion import from the extracellular space to the intracellular space[15]. In general, when a rapid and transient

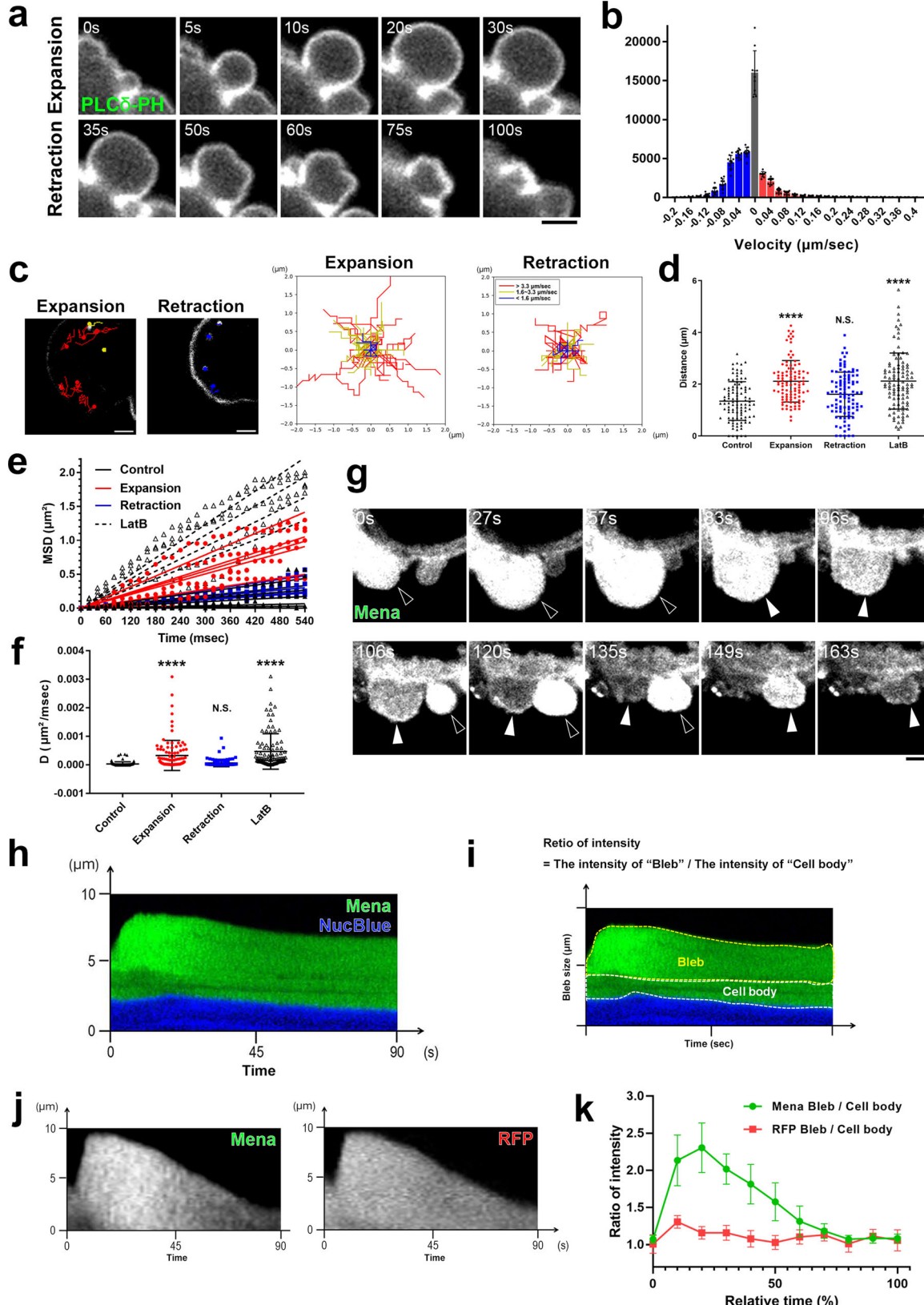

release of calcium ions occurs from the endoplasmic reticulum (ER), the decrease of calcium ion concentration in the ER is sensed by the EF-hand motif of stromal interaction molecules (STIM)[16–18]. STIM proteins then undergo structural changes and rapidly translocate to membrane contact sites that form between the ER and the plasma membrane (PM) where they tether to and

activate the calcium ion-selective Orai1 channels to import calcium ions from the extracellular space[19–21].

First, we performed live imaging observation of an ER marker mCherry-Sec61β to examine the membrane contact sites between the ER and the PM. The membrane contacts sites form only during the bleb expansion (Fig. 4a and Supplementary Movie 5).

**Fig. 1 Highly fluid cytoplasm is formed in the expanding membrane blebs. a**, **b** Membrane blebbing of DLD1 cells expressing GFP-PLCδ-PH. Bin widths of the histogram (**b**) are 0.02 μm/s and the data presented are the means ± the standard deviation (SD) of measurements from 10 independent cells over three independent experiments. **c**–**f** QDs dynamics during bleb expansion and retraction. **c** Example tracks superimposed in situ (left) and centered trajectory maps showing the sum of the frame-to-frame distances over 600 msec (30 frames at 50 Hz; right). Red, yellow, and blue trajectories indicate diffusion velocities >3.3 μm/s, >1.6 μm/s and <3.3 μm/s, and <1.6 μm/s, respectively. See also Supplementary Movie 1. **d** Cumulative distances of QDs motion. **e** Mean square displacement (MSD) analysis of five representative trajectories per condition. **f** Diffusion coefficient (*D*, μm²/msec) was calculated from the slope of the fitted regression line derived by MSD analysis of (**d**). QDs dynamics in the cell body of vehicle- (control) or Latrunculin B (LatB)-treated cells are also shown. *N* = 100 particles from 10 blebs from 10 independent cells per condition. Individual data points are plotted with the means ± SD in **d** and **f**. ****$P < 0.0001$ (One-way ANOVA with Tukey's post-hoc multiple comparison test). **g** Membrane blebbing of DLD1 cells expressing GFP-Mena. Black and white arrowheads show expanding and retracting blebs, respectively. Result shown is representative of five independent experiments. See also Supplementary Movie 2. **h**–**k** Kymograph analysis of GFP-Mena in bleb cytoplasm. **h** Representative kymographs of NucBlue (blue) and GFP-Mena (green) from three independent experiments. Bleb extension is shown on the vertical axis, and time is shown on the horizontal axis. **i** Schematic of the analysis in (**j**, **k**). Fluorescence intensities in "bleb" (yellow) and "cell body" (white) cytoplasm were quantified and expressed as ratios. **j** Representative kymographs of GFP-Mena (left) and the control cytoplasm protein, RFP (right). **k** Fluorescence intensities of GFP-Mena and RFP were quantified as in **i**. Data presented are means ± SD based on the values from five independent experiments. **a** and **g** Indicated times are relative to the first image. Scale bar, 2 μm. Source data are provided as a Source Data file.

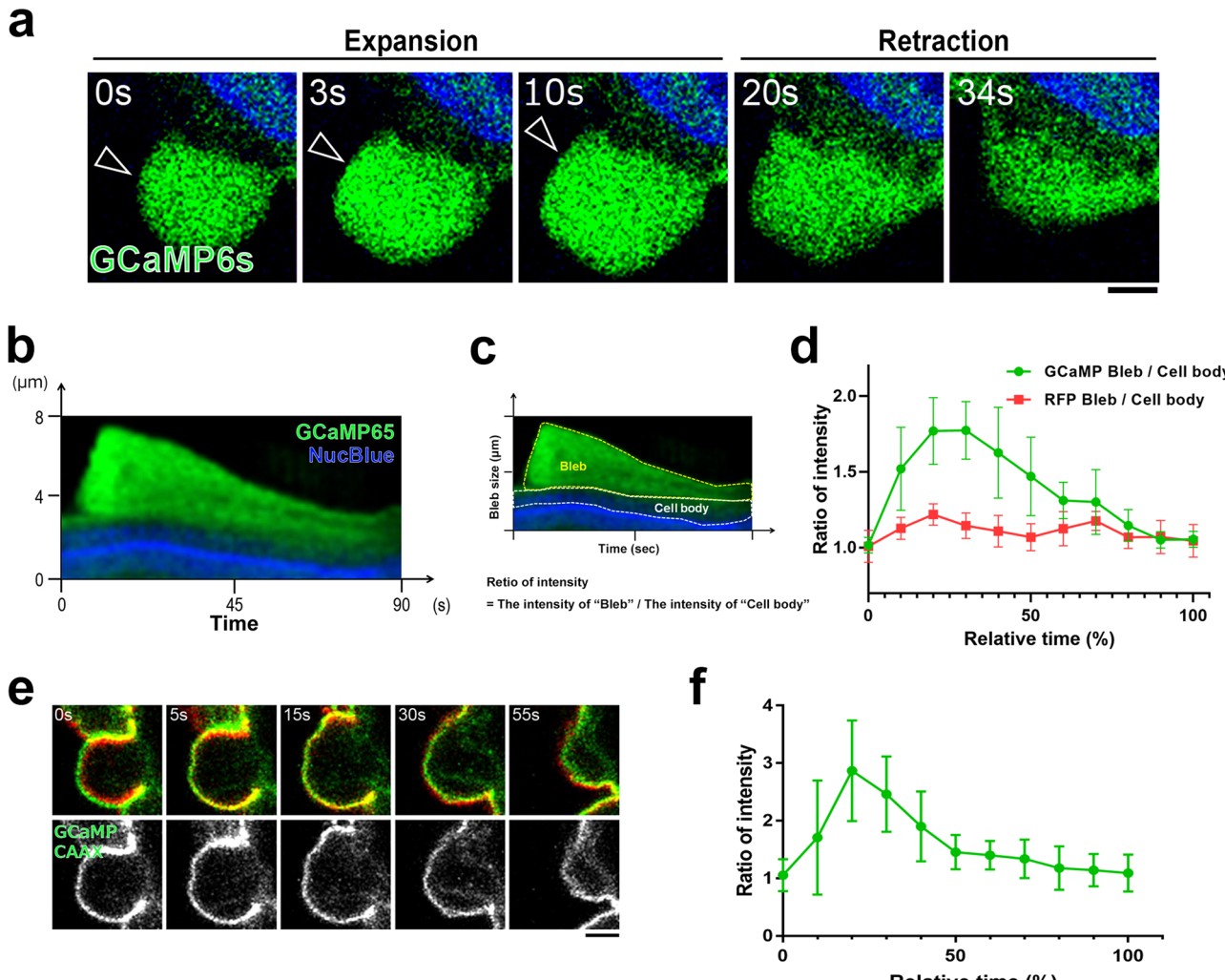

**Fig. 2 The concentration of calcium ion increases in the expanding membrane blebs. a** Cytoplasmic calcium ion concentration was monitored by expressing GCaMP6s. Arrowheads show expanding bleb. Result shown is representative of five independent experiments. See also Supplementary Movie 3. **b**–**d** Kymograph analysis of GCaMP6s in bleb cytoplasm. **b** Representative kymographs of NucBlue (blue) and GCaMP6s (green) from three independent experiments. Bleb extension is shown on the vertical axis, and time is shown on the horizontal axis. **c** Schematic of the analysis in **d**. Fluorescence intensities in "bleb" (yellow) and "cell body" (white) cytoplasm were quantified and expressed as ratios. **d** Fluorescence intensities of GCaMP6s and RFP were quantified as in **c**. **e**, **f** Calcium ion concentration at the juxtamembrane cytoplasm was monitored with the membrane-targeted calcium probe (GCaMP6s-CAAX; **e**). Fluorescence intensities of GCaMP6s-CAAX were normalized to those of mCherry-PLCδ-PH (**f**). Data presented in **d** and **f** are means ± SD based on the values from five independent experiments. **a** and **e** Indicated times are relative to the first image. Scale bar, 2 μm. Source data are provided as a Source Data file.

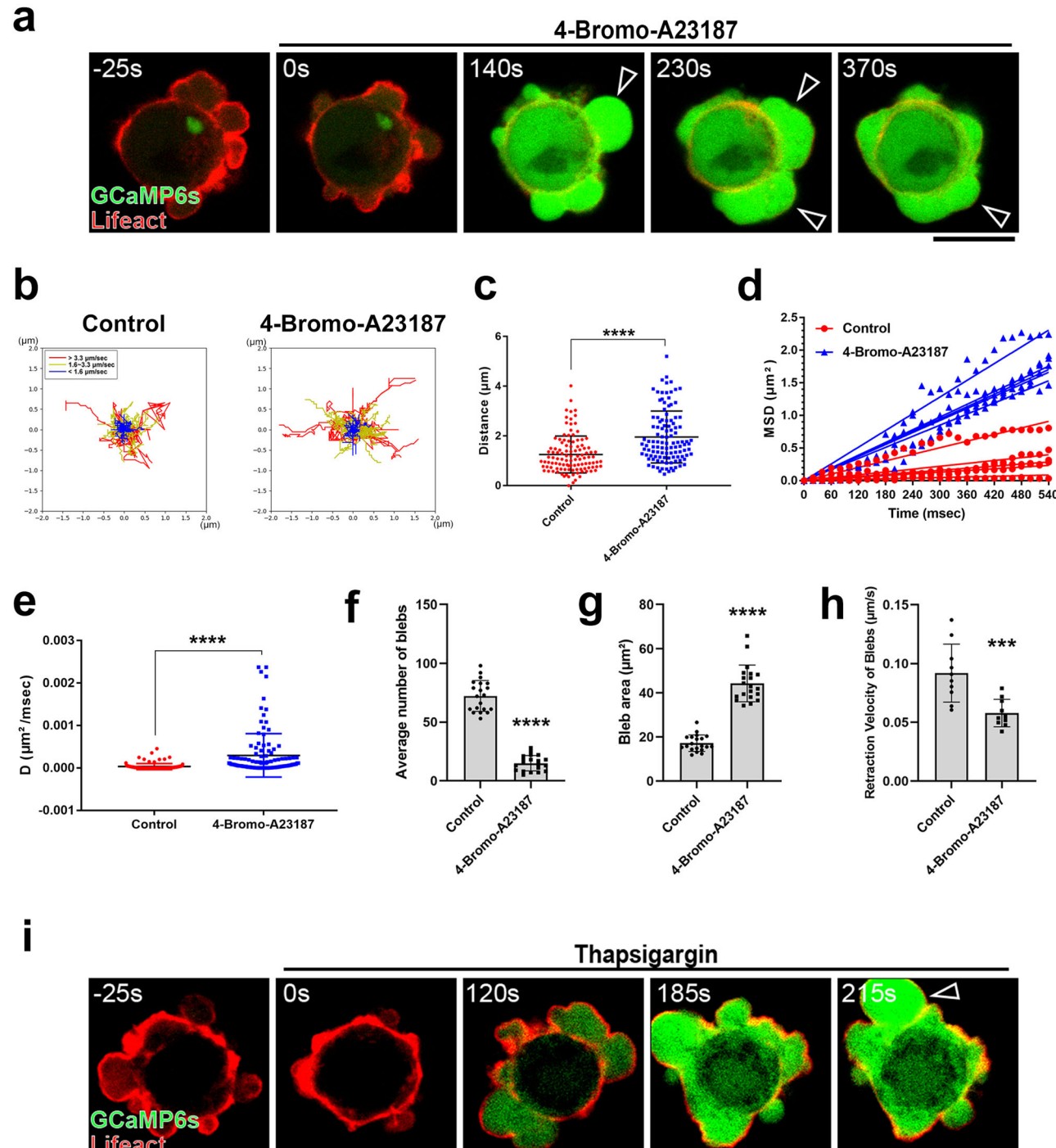

**Fig. 3 Artificially increasing the cytoplasmic calcium ion concentration upregulates cytoplasmic fluidity and exaggerates membrane blebbing. a** Membrane blebbing of DLD1 cells expressing GCaMP6s and Lifeact-RFP treated with the calcium ionophore 4-bromo-A23187 (10 μM). Arrowheads show blebs fusing. Result shown is representative of three independent experiments. See also Supplementary Movie 4. **b–e** QDs dynamics in the cell body of ionophore-treated cells. **b** Centered trajectory maps showing the sum of the frame-to-frame distances over 600 msec (30 frames at 50 Hz; right). Red, yellow, and blue trajectories indicate diffusion velocities >3.3 μm/s, >1.6 μm/s and <3.3 μm/s, and <1.6 μm/s, respectively. **c** Cumulative distances of QDs motion. **d** Mean square displacement (MSD) analysis of five representative trajectories per condition. **e** Diffusion coefficient (D, μm²/msec) was calculated from the slope of the fitted regression line derived by MSD analysis of (**c**). N = 110 particles from 11 blebs from 10 independent cells per condition. **c**, **e** The number (**f**, N = 20 cells), area (**g**, N = 20 blebs) and retraction velocity (**h**, N = 10 blebs) of membrane blebs in vehicle-treated (control) and ionophore-treated (4-bromo-A23187) DLD1 cells over 10 min from three independent experiments. **i** Membrane blebbing of DLD1 cells expressing GCaMP6s and Lifeact-RFP treated with the SERCA inhibitor Thapsigargin (1 μM). Arrowhead shows increased GCaMP6s signal intensity in the bleb. Result shown is representative of three independent experiments. **a** and **i** Indicated times are relative to drug treatment. Scale bar, 10 μm. **c**, **e-h** Individual data points are plotted with the means ± SD. ****P < 0.0001 (Two-sided, unpaired Student's t test). Source data are provided as a Source Data file.

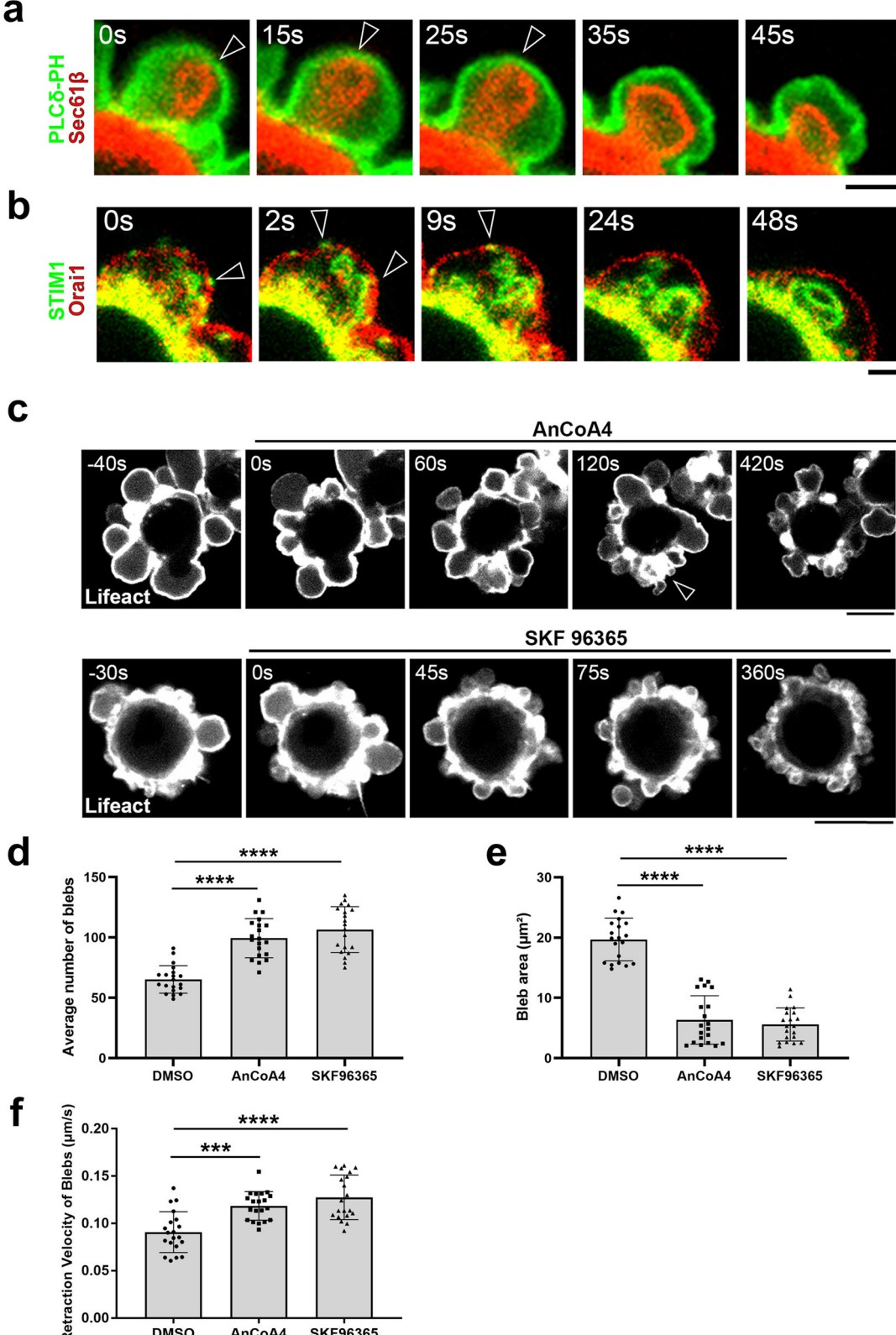

**Fig. 4 The influx of calcium ions via SOCE support expansion of membrane blebs.** Membrane blebbing of DLD1 cells expressing either Sec61β-mCherry and GFP-PLCδ-PH (**a**, Supplementary Movie 5) or Orai1-mCherry and GFP-tagged STIM1 (**b**). Arrowheads show the ER-PM contact sites. Indicated times are relative to the first image. Results shown are representative of three independent experiments. Scale bar, 2 μm. **c–f** DLD1 cells expressing Lifeact-RFP were treated with the SOCE inhibitors AnCoA4 (50 μM) and SKF96365 (10 μM). **c** Representative images from three independent experiments. Indicated times are relative to drug treatment. Scale bar, 10 μm. The number (**d**, $N = 20$ cells), area (**e**, $N = 20$ blebs) and retraction velocity (**f**, $N = 20$ blebs) of membrane blebs in vehicle-treated (control) and drug-treated (4-bromo-A23187) DLD1 cells over 10 min from three independent experiments. Individual data points are plotted with the means ± SD. ****$P < 0.0001$ (One-way ANOVA with Tukey's post-hoc multiple comparison test). Source data are provided as a Source Data file.

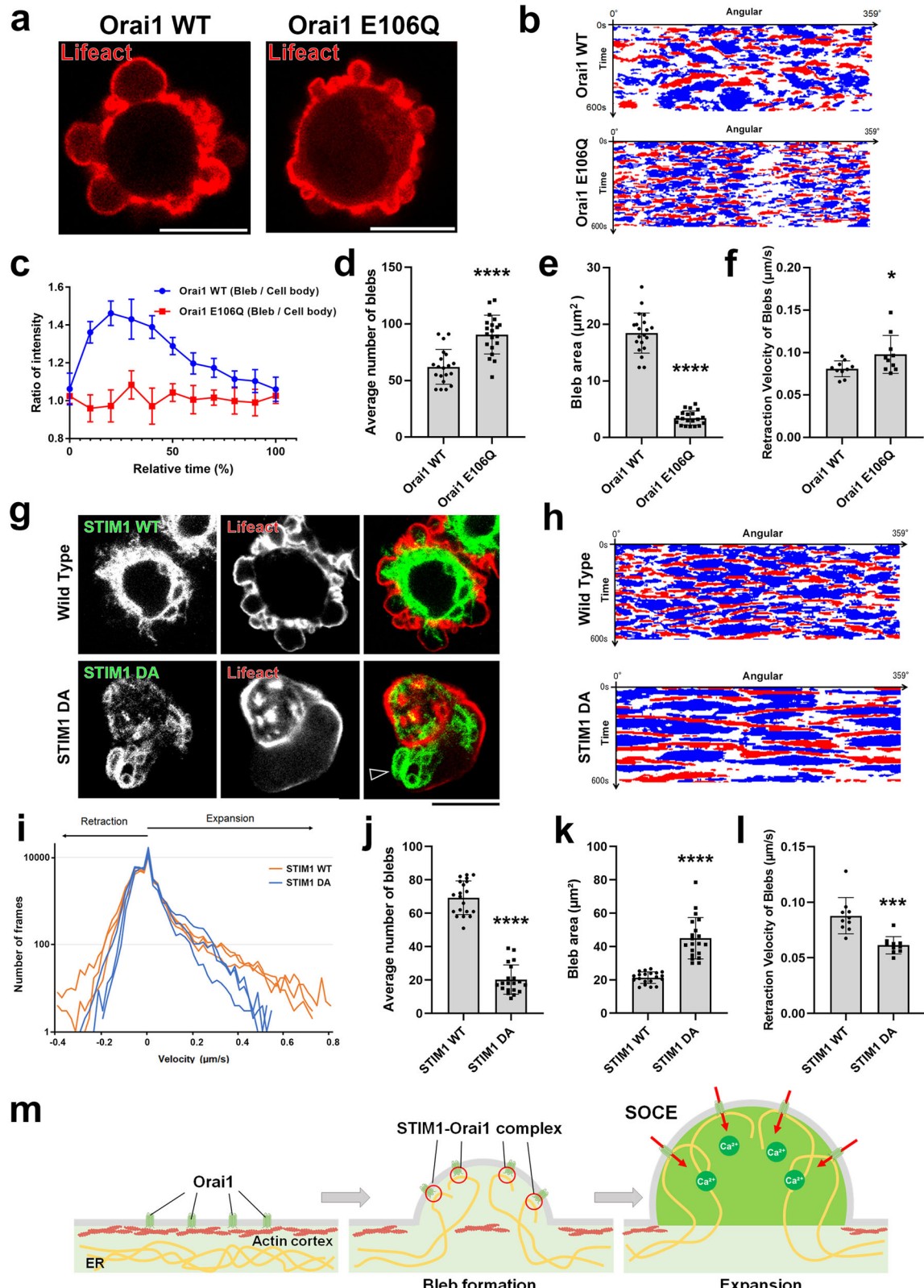

Looking more closely, ER flows into the expanding bleb from gaps of the actin filaments that are remnants of the actin cortex (Supplementary Fig. 4a, b). In addition, STIM1 and Orai1 co-localize only in the expanding bleb (Fig. 4b). Therefore, we examined the necessity of SOCE for bleb expansion by treating cells with the SOCE inhibitors AnCoA4[22] and SKF96365. Both SOCE inhibitors markedly impaired bleb expansion (Fig. 4c–f).

Suppression of SOCE by overexpressing the dominant negative mutant of Orai1 (Orai1E106Q) inhibited the transient rise in calcium ions during expansion phase and shortened the bleb expansion phase, which decreased bleb size (Fig. 5a–f). By contrast, the expansion phase was prolonged, and bleb size increased, when the constitutive active mutant of STIM1 (STIM1 D76A) was expressed (Fig. 5g–l).

**Fig. 5 The formation of STIM1-Orai1 complex promotes the expansion of blebs. a–f** Membrane blebbing in DLD1 cells expressing Lifeact-RFP with either WT or dominant negative (E106Q) Orai1. **a** Representative still images from five independent experiments. **b** Tricolor maps showing angular coordinates along the horizontal axis and time on the vertical axis. Red zones represent expansion, blue zones represent retraction, and white zones represent no movement. Results shown are representative of three independent experiments. **c** Fluorescence intensities of GCaMP6s in "bleb" and "cell body" cytoplasm were quantified in DLD1 cells expressing either WT or E106Q Orai1. The ratio of "bleb" to "cell body" intensities are plotted over time. Data presented are means ± SD based on the values from five independent experiments. The number (**d**, $N = 20$ cells), area (**e**, $N = 20$ blebs), and retraction velocity (**f**, $N = 10$ blebs) of membrane blebs in WT or E106Q Orai1-expressing cells. **g–l** Membrane blebbing in DLD1 cells expressing Lifeact-RFP with either WT or dominant active (DA, D76A) STIM1. **g** Representative still images from three independent experiments. Arrowhead indicate persistent ER-PM contact site. **h** Tricolor maps showing angular coordinates along the horizontal axis and time on the vertical axis. Red zones represent expansion, blue zones represent retraction, and white zones represent no movement. Results shown are representative of three independent experiments. **i** Histograms of bleb expansion and retraction velocities in WT an DA STIM1-expressing cells. Three independent measurements are plotted for each condition. The number (**j**, $N = 20$ cells), area (**k**, $N = 20$ blebs), and retraction velocity (**l**, $N = 10$ blebs) of membrane blebs in WT or DA STIM1-expressing cells. Individual data points are plotted with the means ± SD. **m** Schematic of PM-ER contact site formation in the early stages of bleb formation. Details of the model are described in the text. **a** and **g** Scale bar, 10 μm. **d–f**, **j–l** ****$P < 0.0001$ (Two-sided, unpaired Student's $t$ test). Source data are provided as a Source Data file.

The above observations are summarized in the schematic shown in Fig. 5m. First, blebs are formed by the protrusion of the PM according to intracellular pressure at sites of local defect in the actin cortex. More precisely, during the earliest stages of bleb formation, a patchy defect in the actin cortex is observed, at which point ER-PM contact sites are formed (Supplementary Fig. 4). The ER is propelled into the bleb cytoplasm during the subsequent expansion phase, throughout which the ER-PM contact sites are maintained (Fig. 4a). The formation of the ER-PM contact sites is responsible for the influx of calcium ion from the extracellular space via SOCE, which spurs the increase in the cytoplasmic fluidity of the expanding bleb. Making the cytoplasm less viscous in the expanding bleb is essential for its rapid expansion since treatment of cells with SOCE inhibitors significantly impedes its enlargement (Fig. 4c). Taken together, we concluded that the influx of calcium ion via SOCE is essential to augment the supply of calcium ion to the level necessary to increase cytoplasmic fluidity and drive bleb expansion.

**Activated Ezrin inhibits SOCE by directly binding to Orai1 and promotes retraction of membrane blebs.** We next asked how the influx of calcium ion via SOCE is terminated during bleb retraction. The actin cytoskeleton is rapidly reconstructed beneath the cell membrane as the bleb cycle turns from the expansion phase to the retraction phase. We previously showed that Rnd3 is present in the plasma membrane during bleb expansion and RhoA is distributed in the plasma membrane during bleb retraction[23]. Rnd3 activates p190-RhoGAP to suppress activation of RhoA and inhibit reassembly of actin cortex. Conversely, RhoA promotes actin polymerization and RhoA-ROCK phosphorylates Rnd3 to displace Rnd3 from the plasma membrane of the retracting bleb. In addition, RhoA-ROCK phosphorylates and activates Ezrin, thereby enhancing the association of the plasma membrane with the actin cytoskeleton.

When the bleb starts to retract, E-syt1, a marker protein of the ER-PM contact sites[24,25], and the actin cytoskeleton were distributed exclusive of each other in the plasma membrane (Fig. 6a and Supplementary Movie 6). This finding indicates that ER-PM contact sites are inhibited by the reforming actin cytoskeleton network. Accordingly, treatment of cells with the actin polymerization inhibitor Latrunculin B increased the area of ER-PM contact sites and the cytoplasmic calcium concentration (Fig. 6b, c and Supplementary Movie 7).

In our previous study, we reported that loss of Ezrin impairs the reassembly of the actin cortex in retracting blebs[23]. The morphology of membrane blebs in Ezrin knock out (KO) cells is similar to those in cells expressing constitutive active mutant of STIM1 (Supplementary Fig. 5a–e). Therefore, we investigated the

possibility that Ezrin directly inhibits SOCE in retracting blebs. Intriguingly, the cytoplasmic SOAR domain of STIM1 responsible for the binding to Orai1 was reported to show high sequence similarity to Ezrin/Radixin/Moesin (ERM) domain of Ezrin[26–28]. We hypothesized that Ezrin, which is activated by RhoA-ROCK, binds Orai1 to competitively exclude STIM1 from retracting blebs and terminate SOCE. Immunoprecipitation confirmed the direct binding of Ezrin to Orai1 (Fig. 6d). Notably, co-expression of the constitutive active form of Ezrin (Ezrin T567E) disrupted STIM1-Orai1 interaction in an expression-dependent manner (Fig. 6e, f). We also performed a proximity ligation assay (PLA), which detects protein-protein interaction between proteins within 40–100 nm of each other in situ[29]. PLA showed that the direct interaction between STIM1 and Orai1 occurred only in the plasma membrane of expanding blebs, as no PLA signal was detected in the retracting blebs where Lifeact-signal is present, supporting the idea that Ezrin inhibits interaction between STIM1 and Orai1 (Fig. 6g, h and Supplementary Fig. 6a). We confirmed that the PLA signal between STIM1 and Orai1 was significantly increased when the ER lumenal calcium ion was forcibly depleted by the treatment with thapsigargin, which inhibits the uptake of calcium into ER (Supplementary Fig. 6b, c).

The area of membrane contact sites between the ER and the PM abnormally persisted in Ezrin KO cells or cells expressing constitutive active Rnd3, which inhibits Ezrin activation (Fig. 6i and Supplementary Fig. 5f). Conversely, overexpression of Ezrin T567E, a constitutive active mutant of Ezrin, suppressed the formation of ER-PM contact sites (Fig. 6j).

Since bleb formation was completely suppressed in Ezrin T567E-expressing cells, we wondered if disrupting the actin cortex in these cells could induce bleb formation. In Ezrin T567E-expressing cells, formation of the ER-PM contact sites and the influx of calcium ions via SOCE were suppressed (Supplementary Fig. 7a, b). Accordingly, while Cytochalasin D treatment of these cells enabled bleb formation, those that formed failed to expand, resulting in smaller blebs compared to control cells (Supplementary Fig. 7c–e).

Next, we examined whether the observation that SOCE is suppressed by activated Ezrin is a regulatory mechanism common to other cell systems. When cells were treated with thapsigargin, which depletes the ER of calcium ion and induces SOCE, intracellular calcium ion increased rapidly both in DLD1 cells and in the adherent epithelial cell line, MDCK II. Importantly, thapsigargin-induced SOCE was significantly suppressed in both cells when Ezrin T567E was over-expressed (Supplementary Fig. 7f–i). Taken together, we concluded that RhoA-activated Ezrin not only actuates actin polymerization in the plasma membrane of retracting blebs but also inhibits SOCE to decrease

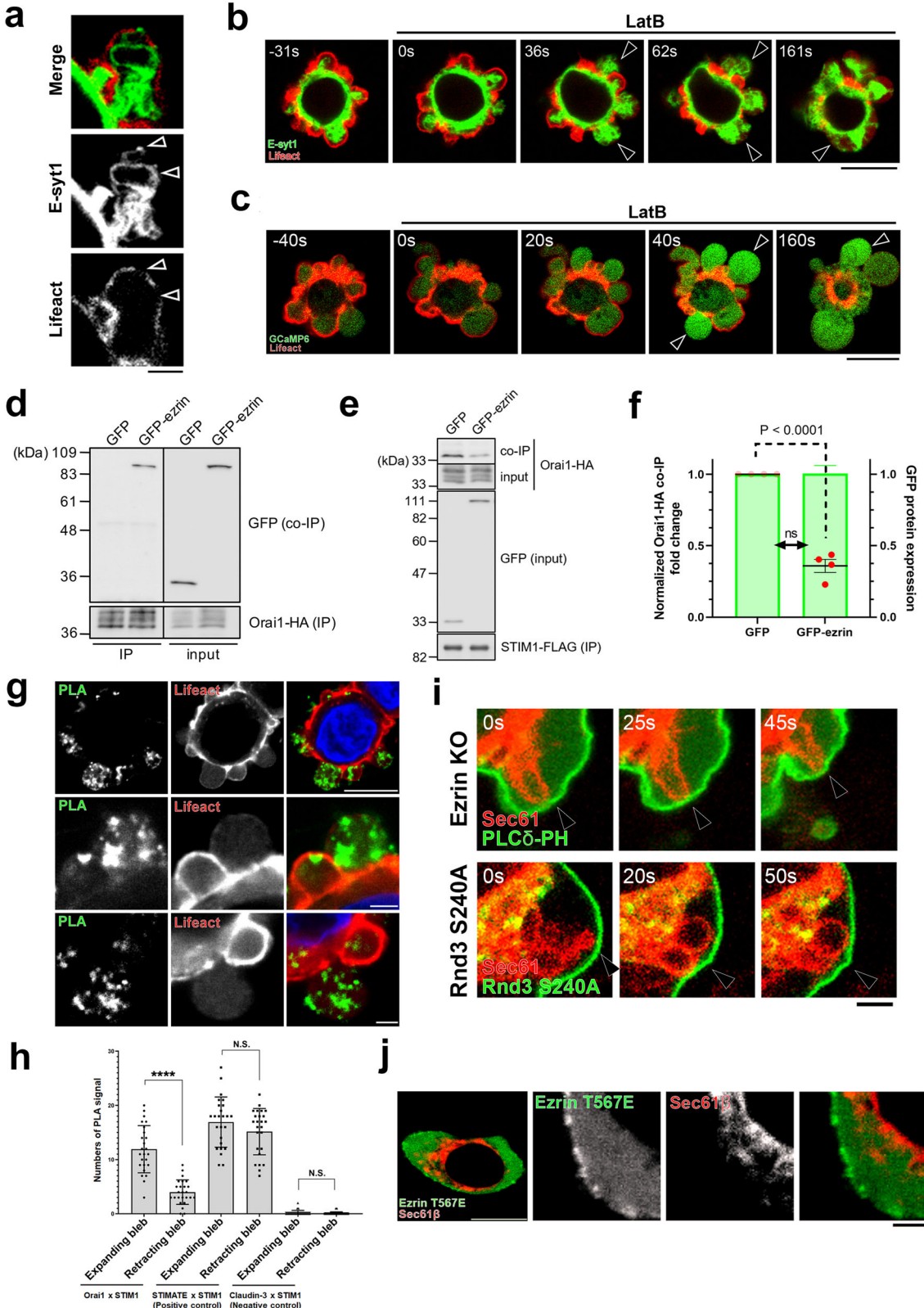

calcium concentration in the bleb cytoplasm by directly binding to Orai1.

**SOCE formation is necessary for the dissemination of amoeboid single cells from cancer spheroid in 3D environment.** Finally, we examined the functional significance of SOCE-regulated cytoplasmic fluidity in bleb-based amoeboid

migration. DLD1 cells overexpressing either the dominant negative form of Orai1 (Orai1E106Q) or the constitutively active form of Ezrin (Ezrin T567E) exhibited markedly reduced migration in the Boyden chamber migration assay (Fig. 7a). Cancer cells can transition between adhesion-based mesenchymal motility and bleb-based amoeboid migration to increase invasiveness. Spheroid culture of the mouse mammary gland

**Fig. 6 Activated Ezrin inhibits SOCE by directly binding to Orai1 and promotes retraction of membrane blebs. a** Representative still images of membrane blebbing in DLD1 cells expressing Lifeact-RFP and GFP-E-Syt1 from five independent experiments. See also Supplementary Movie 6. **b, c** DLD1 cells expressing Lifeact-RFP with either GFP-E-Syt1 (**b**) or GCaMP6s (**c**) were treated with the actin polymerization inhibitor Latrunculin B (LatB, 10 μM). Arrowheads show disrupted actin cortex. Results shown are representative of three independent experiments. See also Supplementary Movie 7. **d–f** Biochemical analyses of ezrin's role in STIM1-Orai1 complex formation. **d** Orai1-HA was co-expressed with either GFP or GFP-ezrin. Inputs and HA immunoprecipitates were immunoblotted with antibodies against HA and GFP. **e** STIM1-FLAG and Orai1-HA were co-expressed with either GFP or GFP-ezrin. Inputs and FLAG immunoprecipitates were immunoblotted with antibodies against FLAG, HA, and GFP. **f** GFP expression levels (shaded in green) and co-precipitated Orai1-HA (individual measurements) were quantified and normalized to GFP-expressing control. Data presented are means ± SD based on the values from four independent experiments. **g, h** Proximity ligation assay (PLA) for in situ detection of Orai1-STIM1 interaction. Representative images from five independent experiments are shown in **g**. **h** Quantification of PLA signals in expanding (low Lifeact intensity) and retracting (high Lifeact intensity) blebs based on data shown in **g** and Supplementary Fig. 6a. STIMATE-STIM1 and Claudin-3-STIM1 are positive and negative controls, respectively. N = 25 independent blebs. Individual data points are plotted with the means ± SD. **i** Membrane blebbing in Ezrin KO cells expressing GFP-PLCδ-PH and Sec61β-mCherry (upper panels) and in WT DLD1 cells expressing constitutive active Rnd3 (S240A) and Sec61β-mCherry (lower panels). Arrowheads indicate persistent ER-PM contact sites. Results shown are representative of three independent experiments. **j** Representative images from three independent experiments of DLD1 cells expressing constitutive active Ezrin (T567E) and Sec61β-mCherry. **a, g** (middle and bottom panels), **i, j** (three right panels) Scale bar, 2 μm. **b, c, g** (top panels), and **j** (left panel) Scale bar, 10 μm. **f, h** ****$P < 0.0001$ (Two-sided, unpaired Student's t test). Times shown are relative to drug treatment (**b, c**) or the first image (**i**). Source data are provided as a Source Data file.

cancer-derived 4T1 cells switch their mode of migration from adherent collective cell migration to bleb-propelled amoeboid cell migration under hypoxic condition (Fig. 7b)[30]. Treatment of the 4T1 cell spheroid with a SOCE inhibitor greatly reduced the number of single cells that showed amoeboid cell migration, suggesting that SOCE is necessary for the induction of amoeboid cell migration (Fig. 7c, d). Importantly, we observed that the ER actively flowed into expanding blebs in amoeboid migrating 4T1 cells as in DLD1 cells (Fig. 7e).

## Discussion

Amoeboid cell movement is observed in various biological contexts such as development[3], wound healing[31], and cancer metastasis[32]. The mechanism of bleb-driven amoeboid locomotion was recently proposed[6]. According to this model, contractility of the actomyosin cortex increases the hydrostatic pressure inside the cell, which leads to a regional breach of the cell cortex and initiates polarized bleb formation. This is followed by the flow of cytoplasm into the protruding bleb to sustain bleb formation in the direction of locomotion. However, a key element is unaccounted for in this general description; that is, cytoplasmic fluidity. If the cortex strength and configuration of cytoplasmic fluidity are not controlled synchronously, the cell cannot migrate in the desired direction. Specifically, the cytoplasmic fluidity must be mechanistically coupled to bleb dynamics in order to produce localized cytoplasmic flow near-simultaneous to protrusion formation. Thus, the question that must be addressed is how the interaction between the cell membrane and cytoskeleton and the fluidity of cytoplasm are cooperatively regulated.

Actin is one of the most abundant components of the cytosol and polymerized actin stiffens the cytoplasm[33]. The state of actin is controlled by actin-binding proteins and calcium ion is a prominent regulator of these proteins that can exert multiple effects on the structure and dynamics of the actin cytoskeleton. It was previously demonstrated in vitro that calcium ions are integral to cytosolic sol-gel conversion[13]. The increase of calcium ion severs F-actin and depolymerizes it by activation of gelsolin[34] and INF2[35] and upregulates the cytoplasmic fluidity. On the one hand, the increase of calucium ion can also enhance the contractility of acto-myosin cortex. In the present study, we revealed that the concentration of calcium ion was up-regulated only in the expanding bleb cytoplasm. Since the plasma membrane of expanding blebs is devoid of the acto-myosin cortex—and since the calcium ion concentration in the cell body is unchanged at a steady state level—we propose that the increase in calcium ion

during the expansion phase drives bleb dynamics by explicitly increasing the cytoplasmic fluidity in the expanding bleb.

As for the molecular mechanism involved in the regulation of cytoplasmic fluidity during bleb cycle, we revealed the existence of mechanisms to actively increase the fluidity of the cytoplasm by specifically up-regulating calcium ions in the cytoplasm of expanding blebs. However, the possibility remains that the severed actin filaments emanating from the disintegrating actin cortex functions as a sieve to limit particulate infusion into the expanding bleb. These possibilities need to be explored in more detail in future studies.

We show that ER-PM contact sites are formed, and influx of calcium ion via SOCE, is facilitated during the expansion phase of the bleb. This observation is in good agreement with previous reports that the actin cortex acts as a physical barrier that impedes SOCE[36,37]. In our previous study, we showed that Rnd3 is present at plasma membrane of blebs during the expansion phase and suppresses actin polymerization during the expansion phase[23]. Indeed, the suppression of actin cortex formation by the expression of constitutively active Rnd3 expands EM-PM contacts and promotes the SOCE (Fig. 6i). It was also recently reported that cortical actin is decreased when SOCE is initiated because the ER stress sensor PKR-like endoplasmic reticulum kinase (PERK) sequesters an actin cross linker protein, Filamin-A, at the ER and changes the spatial organization of F-actin[38]. Therefore, there appears to be a positive feedback loop between the reduction of actin cortex and SOCE. Furthermore, it was reported that bleb formation is enhanced in Filamin-A deficient cells[39]. Thus, PERK-Filamin-A axis may support the robust calcium ion influx during the expansion phase of bleb dynamics.

In conclusion, our present study revealed that the expansion and retraction of membrane bleb are orchestrated by the regulation of cytoplasmic calcium concentration by STIM-Orai1 signaling and by the regulation of actin polymerization by RhoA-Rnd3 antagonism (Fig. 7f). Activation of STIM1-Orai1 pathway is reported to promote invasion of malignant melanoma[40]. This study provides the rational theoretical basis for how inhibitors targeting the STIM1-Orai1 pathway suppress amoeboid-like movements of cancer cells and consequently inhibit invasion.

## Methods

**Reagents**. DLD1 cells, MDCK II cells, HEK293 cells, and 4T1 cells were purchased from ATCC. DLD1 cells, MDCK II and HEK293 cells were grown in DMEM supplemented with 10% (vol/vol) fetal calf serum (FCS) (Sigma). 4T1 cells were cultured in RPMI1640 supplemented with 10% (vol/vol) fetal calf serum (FCS) (Sigma). DLD1 cells were treated with 50 μM AnCoA4 (Calbiochem) and 10 μM SKF96365 (Tocris Bioscience) to inhibit SOCE activity. Latrunculin B (Abcam) was

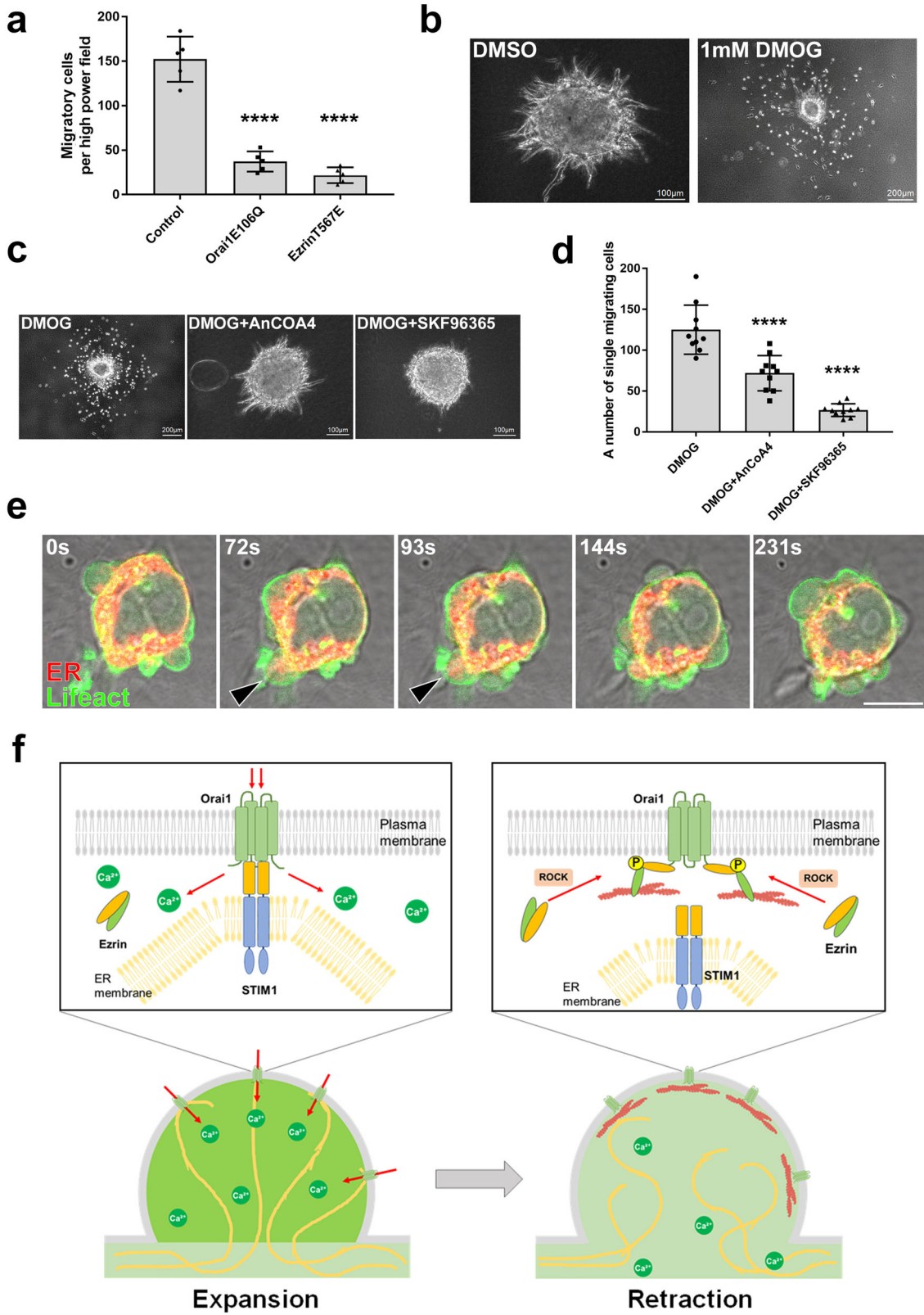

used at 5 μM. Jasplakinolide (Invitrogen) was used at 10 μM. 4-bromo-A23187 (Sigma-Aldrich) was used at 10 μM. Thapsigargin (FUJIFILM Wako Pure Chemical Corporation, Osaka, Japan) was used at 1 μM. The following primary antibodies were used for immunoblotting: Mouse anti DYKDDDDK tag, monoclonal antibody (#014-22383, FUJIFILM Wako Pure Chemical, 1:1000), Rabbit anti-HA-tag antibody (#561, MBL, Nagoya, Japan, 1:1000) and Mouse anti-GFP antibody (#11814460001, Roche, 1:1000). Secondary antibodies were as follows: Goat

Anti-Rabbit IgG-HRP (#4030-05, Southern Biotech, 1:1000); Goat anti-Mouse IgG-heavy and light chain cross-adsorbed Antibody HRP Conjugate (#A90-516P, Bethyl Laboratories, 1:1000). Alexa Fluor 555 Phalloidin (#A34055, 1:100) was purchased from Thermo Fisher Scientific. Following expression vectors were purchased from Addgene; pGP-CMV-GCaMP6s-CAAX (Addgene No.52228), Sec61β-mCherry (Addgene No. 49155), pGP-CMV-GCaMP6s (Addgene No.40753), EGFP-E-Syt1 (Addgene No.66830), GFP-PIP5K gamma (Addgene No.

**Fig. 7 SOCE formation is necessary for the dissemination of amoeboid single cells from cancer spheroid in 3D environment. a–e** Role of SOCE in cell migration. **a** Transwell migration assays using non-transfected DLD1 cells and cells expressing either dominant negative Orai1 (E106Q) or constitutive active Ezrin (T567E). Individual data points are plotted with the means ± SD based on the values from $N = 5$ independent experiments. **b** 4T1 cell spheroids embedded in type I collagen were treated with DMSO (left panel) or DMOG (1 mM), an inhibitor of prolyl hydroxylase, to activate HIF-1, (right panel). Scale bar, 100 μm (left panel) or 200 μm (right panel). **c** Representative 4T1 cell spheroid embedded in type I collagen was treated with DMOG only (1 mM, left panel), or with a combination of DMOG and SOCE inhibitors AnCoA4 (50 μM, center panel) or SKF96365 (10 μM, right panel) for 72 h. Scale bar, 200 μm (left panel) or 100 μm (center and right panels). **d** Number of single cells migrating away from the spheroid were quantified based on the experiment shown in **c**. $N = 10$ independent spheroids per condition from three independent experiments. Individual data points are plotted with the means ± SD. **e** Representative amoeboid migration of 4T1 cells expressing Lifeact-GFP and stained with ER-tracker (red). after spheroid treatment with DMOG as in **b** from three independent experiments. Arrowheads indicate regions where the ER flows into blebs. Times shown are relative to the first image. Scale bar, 10 μm. **f** Schematic showing the proposed mechanism of SOCE-dependent cytoplasm solation. In the expansion phase of membrane blebbing, the influx of calcium ions via SOCE inhibits actin polymerization and promotes solation of the cytoplasm, which supports rapid bleb expansion. By contrast, during the retraction phase, activated Ezrin inhibits SOCE by directly binding to Orai1, allowing the reassembly of the actin cytoskeleton by decreasing the calcium ion concentration in the cytoplasm of the bleb. **a, d** ****$P < 0.0001$ (One-way ANOVA with Tukey's post-hoc multiple comparison test). Source data are provided as a Source Data file.

22299). cDNAs encoding full-length STIM1, Orai1, Mena, VASP, STIMATE, and MRLC1 were amplified by RT-PCR, fused to the sequence encoding EGFP, Scarlet or HA, and ligated into the pCAGGS-neo vector. Primers for RT-PCR are included in Supplementary Table 1. Expression vector of GFP-Rnd3 S240A mutant was kindly provided by Dr. A. S. Yap (University of Queensland, St. Lucia, Queensland, Australia). Ezrin-KO DLD1 cells were characterized previously[23].

**Live Imaging**. Fluorescence imaging was performed using a 63×oil-immersion objective on an inverted microscope (LSM700; Carl Zeiss MicroImaging) interfaced to a laser-scanning confocal microscope equipped with a heating stage heated to 37 °C. Nuclei was stained using NucBlue Live Ready Probe Reagent (Invitrogen). Images were captured on a device camera and acquired using ZEN2012 software (LSM700; Carl Zeiss Micro Imaging). 3D fluorescence imaging was performed using a ×100 oil-immersion objective on an inverted microscope (IX83; Olympus corporation) interfaced to a spinning-disk confocal microscopy (Dragonfly200; OXFORD Instruments) equipped with a heating stage heated to 37 °C. Images were captured on a device camera and acquired using Fusion software (Dragonfly200; OXFORD Instruments). Images were acquired at 488 nm for GFP-tagged proteins or at 555 nm for RFP or mCherry-tagged proteins. Each imaging video frame is a 8-bit grayscale image, and the frame interval is indicated in the supplementary movie legends. The movie captures a single cell. 3D reconstructions were performed using Imaris software (Bitplane). Quantitative analysis of the fluorescent intensity was performed using ImageJ/Fiji.

**Immunoprecipitation**. Cells were washed with PBS and lysed with IP buffer (20 mM Tris-HCl [pH 7.5], 150 mM NaCl, 1% Triton X-100, and protease inhibitors). Lysates were incubated with either 1 μg anti-HA mAb conjugated to Protein G Sepharose (GE Healthcare) or 5 μl of anti-DYKDDDDK mAb beads (Wako Pure Chemical Industries) for 2 h. Beads were washed with IP buffer and bound proteins were eluted in SDS sample buffer. Aliquots of the lysate and eluate were immunoblotted with anti-FLAG antibody (1:1000), anti-HA antibody (1:1000) and anti-GFP antibody (1:1000).

**Introduction of quantum dots into cells**. QDs were introduced into DLD1 cells by electroporation using a NEPA21 Super Electroporator (NEPAGENE, Tokyo, Japan). In all, 1 μl QDs in 1 μl suspension buffer (Qtracker 605 Cell Labeling Kits (Invitrogen)) were suspended in 200 μl Opti-MEM (Invitrogen) and mixed with $1 \times 10^6$ DLD1 cells in a green cuvette with a 1-mm gap (NEPAGENE) and set to Electroporator. Poring pulse was optimized between 125 V and 275 V for 2.5 or 5.0 msec pulse length twice with a 50 msec interval between the pulses and 10% decay rate with + polarity. The transfer pulse condition was five pulses at 20 V for 50 msec pulse length with 50 msec interval between the pulses and 40% decay rate with ±polarity. After electroporation, cells were recovered in serum-containing DMEM and plated in a glass-bottom dish. Cells were imaged after 1 h incubation by using a ×100 oil-immersion objective on an inverted microscope (IX83; Olympus corporation) interfaced to a spinning-disk confocal microscopy (Dragonfly200; OXFORD Instruments) equipped with a heating stage set to 37 °C. Images were captured every 20 msec (50 Hz) on a device camera and acquired using Fusion software (Dragonfly200; OXFORD Instruments).

**Proximity ligation assay**. DLD1 cells expressing HA-tagged Orai1, FLAG-tagged STIM1, and Lifeact-tdTomato were cultured on coverslips and fixed with 2% paraformaldehyde prepared in PBS for 15 min at 37 °C. These cells were treated with 100 μg/mL digitonin prepared in PBS for 15 min, and washed with PBS three times. Fixed cells were blocked by incubation with 5% BSA prepared in PBS for 30 min at 37 °C. Cells were incubated at 37 °C for 1 h with the primary antibody,

rabbit anti-HA antibody (1:100), mouse anti-FLAG antibody (1:100), and Rabbit anti Claudin-3 polyclonal antibody (#34-1700, Invitrogen, 1:100). The PLA reaction was detected with the Duolink In Situ PLA kit (Sigma-Aldrich) according to the manufacturer's instructions.

**Transwell migration assay**. Transwell migration assays were performed using Transwell filters (8-μm pore diameter; Corning Costar) coated with 1.0 mg/ml type I collagen (KOKEN). Membrane filter inserts were precoated with 100 μL of 1.0 mg/ml type I collagen solution in each well before seeding cells. In all, $2 \times 10^5$ cells were seeded into the upper chamber, which contained the same medium as the lower chamber. After migration for 6 h, cells on the underside of the Transwell filter were stained with DAPI and the number of cells per high-power field was counted.

**Spheroid culture in 3D collagen**. Multicellular spheroids were generated by using the hanging-drop assay. Spheroids were generated by hanging-drop culture (500 cells/25 μl droplet) and embedded in 4 mg/ml native collagen type I by incubating at 37 °C for 10 min. Embedded spheroids were cultured for 72 h in the presence of either 1 mM Dimethyloxaloylglycine (DMOG) (Tokyo Chemical Industry) or 0.1% (vol/vol) DMSO.

**Statistical analysis**. GraphPad Prism 8.4.1 (GraphPad Software) was used to graph data and to perform statistical analyses. One-way ANOVA, unpaired $t$-test or multiple $t$ tests was performed as appropriate to compare means. Normality of data distribution was assessed by D'Agostino & Pearson's test. Error bars are SD and $P$ values are notated in the figure legends.

**Visualization and quantitative analysis of membrane bleb dynamics**. We previously described the method of image analysis for visualization of membrane bleb dynamics in detail[23]. A tricolor map visualizes the membrane bleb dynamics more directly. An expanding bleb is represented in red and a retracting bleb is represented in blue. If a part of the cell contour is effectively stationary, the corresponding area is represented in white.

**Tracking of particle trajectories**. Tracking of QDs were performed using software developed in-house according to the following parameters. First, signals were enhanced by implementation of the Laplacian-of-Gaussian (LoG) filter then detected by the "Find Maxima" detection algorithm in ImageJ at every frame $t$. Next, point-to-point correspondence between signals in sequential frames were determined using the stable marriage algorithm to form multiple particle trajectories between $t$ and $t + 1$. Finally, particle trajectories of duration <10 frames were discarded as noise.

**Reporting summary**. Further information on research design is available in the Nature Research Reporting Summary linked to this article.

## Data availability
Data supporting the findings of this manuscript are available from the corresponding author upon reasonable request. A reporting summary for this Article is available as a Supplementary Information file. Source data are provided with this paper.

## Code availability
The codes generated during this study are available at the Github repository (https://github.com/uchidalab/bleb-analysis).

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

## Acknowledgements
We thank all members of the Ikenouchi laboratory (Department of Biology, Faculty of Sciences, Kyushu University) for helpful discussions. This work was supported by JSPS KAKENHI (JP19H04968 (J.I.), JP19H03227 (J.I.), JP17H06012 (J.I.), JP19K06640 (M.K.), JP16H06280 (S.U.), JP17K19402 (S.U.), and JP17J00242 (K.A.)), AMED-PRIME (15664862), JST-PRESTO (JPMJPR12A4), grants from the MSD Life Science Foundation, the Takeda Science Foundation, the Uehara Memorial Foundation, the Nakatani Foundation, and a JSPS Research Fellowship for Young Scientists (DC1) (K.A.).

## Author contributions
K.A. performed most of experiments and analyzed the data. S.H., K.K., and S.U. performed quantitative image analysis. K.M. performed some experiments. K.A. and J.I. designed research and wrote the paper.

## Competing interests
Authors declare no competing interests.
