## [Peer Review File · Nature Communications]

Reviewer #1 (Remarks to the Author):

The authors present work that approaches a very significant area of investigation—what controls the formation and retraction of blebbing of membranes on cells. This has implications in the formation of lamellipodia and invadopodia protrusions from cells that mediate cell migration and the invasiveness of cancer cells into the cellular matrix during metastasis. The work examines the premise that calcium entry signals mediated by STIM and Orai mediates this cellular control, which would be very interesting.

Unfortunately, the paper is superficial, with very preliminary and inconclusive data, and written and assembled in a way that makes little sense. The experiments are poorly designed and executed and the logic behind some of the reasoning and conclusions of the paper are very sloppy. This is a shame, since the premise of the paper seemed so significant and the authors seem to have published some good papers recently.

There really are too many problems with this manuscript even to describe, so the following comments just list some of the more obvious concerns:

1. There was no detail on the statistical analysis and quantitation of the live cell imaging experiments (Fig. 1B, 1C, 1F, 1H, 2C, 2E, 2G-2J, 3E-3H, etc). It was not clear how many blebs per cell and how many cells were analyzed, or how many time the experiments were repeated. Was the difference meaningful? Are these data average of multiple cells/blebs or representative data from a single cell/bleb? Without these critical details and appropriate statistical analysis, it is impossible to determine the robustness of the findings and conclusions in the manuscript.

2. It was indicated that screening was performed to search for proteins specifically enriched in the expanding blebs, (page 2 bottom three lines). However, no details were provided anywhere in the manuscript regarding how the screening was performed. What was the nature and composition of the library that they used for the screening? Is this a whole genome library or a more focused library for only actin binding proteins?

3. The logic that A23187 the experiment indicates that restricted up-regulation of calcium concentration in the cytoplasm triggers gel-sol transition of cytoplasm required for bleb expansion, is very difficult, if not impossible to follow. The gel-sol transition was never even measured! And A23187 has not been used in calcium experiments for years since it is fluorescent.

4. The co-IP experiments with overexpressed STIM1 and overexpressed Ezrin are somewhat amateurish and not very convincing. The same with the imaging of ER-PM junctions. At the very minimum a FRET study should have been undertaken and carefully analyzed/compared between expanding and retracting blebs.

5. Labels for x- and y- axis are missing in many panels.

As stated above, these are just some of the problem and not necessarily the major ones. There are many other problems with paper that together would prevent it from being acceptable in any other journal, let alone a high-tier journal.

Reviewer #2 (Remarks to the Author):

This paper by Aoki et al examines the role of Calcium transients in gelation-solation of the cytoplasm. They identify a pathway responsible for elevating calcium in the bleb during growth to solate the cytoplasm and one during retraction to reduce calcium to gelate the cytoplasm. The authors have generated some very nice work about the molecular control of nucleation and contractility in the past and this follows on from it. The topic is interesting and gelation/solation hasn't been examined seriously for a long time, so this is a welcome fresh new look that would clearly fit in Nature Communications.

As it stands, I have several serious conceptual and technical issues with the paper which make me recommend major revisions at least. Some of these issues can be dealt with by rewriting the paper with a slightly different angle, others will need quantification and experiments.

Conceptual issues:

-The authors state in the first line of the abstract: "the cytoplasm in mammalian cells is thought to be homogeneous". I disagree with that statement. I think it is generally accepted that the cytoplasm has both a liquid phase (the cytosol) and a solid phase. When the cell contracts, this increases pressure in the fluid phase and this can lead to detachment of the membrane from the cortex. So, although a calcium influx can lead to increased contraction, it doesn't necessarily have to solate the cytoplasm, it can just squeeze the cytosol out of the cell body to form a bleb somewhere.

-I thinking the concept of STIM-Ora1 being involved in blebbing is interesting. What I think is much more difficult is to show that calcium mediated solation causes blebbing and decrease in calcium causes cortex assembly and gelation of the cytoplasm. Most of the work in the paper indicates a role for STIM-Ora1 in regulating calcium in blebs but how solation/gelation participates is not well-substantiated. For example, when a bleb forms, the membrane detaches from the cortex and the cortex remains at the base of the growing bleb for a few seconds (see Zatulovskiy E et al, JCB, 2014 for example). Because the cortex remains for a few seconds (in fact most of the growth), an alternative explanation is that large macromolecules/organelles are excluded from the bleb because of sieving by the cortex. After a while, the cortex disassembles, probably because actin nucleators are membrane linked, and the barrier preventing macromolecules/organelles from entering into the bleb is removed. Once the barrier is removed, a change in cytoplasmic properties will take place because organelles, macromolecules etc can penetrate into the bleb. In fact, this phenomenon can also be observed by looking at the refractive index of blebs: when they grow they are very transparent and phase clear; while when they retract they are darker. ER probably penetrates into the blebs because it is often found very closely associated to the cortex/membrane and because its thin tube-like structure allows it to deform and get dragged into the bleb by the flow of cytosol. Currently, this sieving hypothesis is not at all considered and not shown incorrect.

-In general, my reading of the paper is that: 1) an initial calcium influx may lead to the initial contraction, 2) continued influx means that the bleb appears to have more calcium, 3) excess calcium is cleared by STIM-Ora1. I think that showing the gelation/solation part convincingly would require a lot more work and careful consideration of the sieving hypothesis. In general, I find the experiments to support the sol-gel hypothesis unspecific: latrunculin treatment is a very blunt tool.

Technical issues:

Major:

- The authors show that the fluorescence intensity of GCamp6 is higher in the bleb than in the rest of the cell. This could either be due to increased calcium in the bleb or it could be because the bleb interior has less excluded volume because there are no organelles or macromolecules when it grows. In movies, examining cytoplasmic GFP, it is not uncommon to get the impression there is more GFP in the blebs than in the cell body because of these effects. In fact, you can see an increase in the fluorescence intensity of RFP on S1E. To examine this carefully, the authors should normalize

fluorescence intensity to that in the cell body and plot the evolution of RFP fluorescence through bleb. The authors should then compare the fluorescence increase in GCamp (similarly normalized) to the fluorescence change for RFP in the blebs (like they have done for VASP/PIP5Kgamma – except that they should normalize to the fluorescence in the cell body). This would clearly exclude the possibility of an artefact due to differences in excluded volume in the cell body vs the bleb and it would convincingly show that we are looking at increased Ca²⁺. From the manuscript figures, it looks like they have done the experiments and they would just need to quantify this.

-I have a similar comment about retraction: because macromolecules and organelles can penetrate in the bleb once the old cortex at the base has been removed, the excluded volume will increase and the fluorescence of a cytoplasmic RFP would decrease. So again, comparing GCamp fluorescence change to mCherry fluorescence change would be good.

-Can the authors observe increases in GCamp fluorescence in the cell body? It would be surprising that Ca²⁺ only increases in the bleb but not in the cell body. Surely there is no barrier stopping calcium from diffusing into the cell body from the bleb.

-On Fig 1D, the bleb indicated by an arrow shows the phenomenon convincingly (i.e. high Mena during expansion, low Mena during retraction) but the one just above it does: it has low Mena fluorescence then this increases as it retracts.

-I find the effect of A23187 difficult to judge since it first makes large blebs but eventually floods the system with calcium and stops blebbing. The blebs appear homogeneously fluorescent in GCamp, yet retraction seems to still proceed with no problem for a while.

-I also find the effect of Ezrin T567E a bit difficult to judge because the cells don't bleb. Can the authors trigger some blebs with a low dose of cytochalasin?

Minor:

-In Fig 1C, the authors should quantify and compare the Mean square displacement of the particles.

-Fig 2H, 3F need to be quantified in some way to allow for the comparisons of the two conditions to be reliably made.

-Fig 2D, the CAAX and the GCamp signal appear to switch polarity in the overlays. We have green distal and red proximal in the first panel but red distal and green proximal in the last. Why do these swap position? In 2E, the authors should show the whole time course from nucleation to retraction.

Reviewer #3 (Remarks to the Author):

This manuscript reports on a tight linkage between the Stim-Orai pathway of store-operated Ca²⁺ entry and the process of membrane blebbing in DLD1 cancer cells. As membrane blebbing is an essential event in amoeboid migration, the authors address a timely and important aspect of Stim-Orai pathophysiology. It is convincingly demonstrated that bleb expansion and retraction is associated with local changes in cytosolic Ca²⁺ within the blebs and involves a Ca²⁺ entry mechanism. Using standard pharmacological tools and expression of a constitutively active Stim1 construct, the authors provide evidence for involvement of Orai-mediated Ca²⁺ entry and for negative regulation of Stim-Orai interaction by activated ezrin. The authors propose that ezrin is critically involved in both reassembly of cortical actin during bleb retraction and suppression of Stim-Orai-mediated Ca²⁺ signaling by competitive interference with Stim-Orai communication. Especially this latter conclusions appear largely speculative and premature. In general, the experimental approaches used to probe Orai signaling are barely adequate. This part of the study needs convincing complementation by more refined experiments as outlined below. Moreover, most results are insufficiently documented, especially statistical information is incomplete. More comprehensive

description of the experiments and illustrations is required.

Specific comments:

1) Orai-mediated bleb cycling in DLD1:

The authors conclude that the proposed Orai-mediated control of bleb cycling may be a key mechanism in cancer cell migration and invasion. Although the linkage between blebbing and invasion is well established, the authors need to show that modulation of Orai function inhibits not only bleb formation but also Ca^{2+} levels within blebs and migration? For this purpose specific genetic manipulation of Orai function such as expression of a dominant negative Orai mutant (E106Q; see also below) are should be used instead of standard, barely specific inhibitors. Also the effects of constitutively active ezrin on bleb Ca^{2+} levels and migration need to be analyzed.

2) Orai-mediated Ca^{2+} signaling and solation in expanding blebs:

In the abstract it is stated that “the solation of the cytoplasm is caused by a sharp rise in the calcium concentration”. I don't see evidence for such causal relation. For this statement one needs to show that fluidity of the bleb cytoplasm (GFP-FRMD4A assay) is indeed affected by modulation Ca^{2+} entry into the bleb. Solation is certainly required for bleb expansion, but, at this stage, one may only conclude that this process is associated with increased Ca^{2+} in the bleb. Moreover, the DLD1 cells appear to display a significant level of constitutive blebbing. Accordingly, in Figure 2A, the experiment starts at time 0 with an almost maximally expanded bleb that shows Ca^{2+} levels probably way above that in the rest of the cytosol. In Figure 2A, this expansion and Ca^{2+} levels then increases moderately for about 10s. In Figure 2C, the time course of Ca^{2+} -sensitive fluorescence in the bleb is apparently compared to the fluorescence of NucBlue and labeled with „cytosol“. This is misleading since one would expect a comparison with the cytosolic Ca^{2+} outside the bleb reported by GCaMP6s. How would a comparison of cytosolic Ca^{2+} within and outside the bleb look? What is the statistics for these experiments. For how many cells/blebs are the displayed images in fact representative? In general, information on experimental statistics is incomplete throughout the paper.

The presented images give the impression that constitutive bleb cycling is associated with a constantly elevated Ca^{2+} level in the bleb. Based on what is known about Stim1-Orai1 function, it is difficult to conceive that this excessive, tonic Ca^{2+} elevation is mediated by Orai as this influx typically shows Ca^{2+} -mediated negative feedback regulation. It appears required to show that intra bleb Ca^{2+} is reduced by specific suppression of Orai function. This can be achieved by either overexpressing just Orai1 (to disrupt Stim-Orai stoichiometry) or by a dnOrai (E106Q).

3) Stim-Orai is accumulated with PM-ER junctions in blebs:

Figure 3 aims to demonstrate that Stim-Orai complexes (punctae) are formed in an expanding bleb. Unfortunately, these illustrations are all but convincing. Again, what is the statistics on these experiments? The Images illustrate that the ER appear to detach from the PM in the retracting bleb. Figure 2B also shows that there is abundant colocalization and probably interaction of Stim and Orai in the basal area of the bleb, while only a few “points” of co-localization are highlighted (arrows) by the authors in the bleb membrane. More convincing would be to demonstrate interaction by FRET. A role of Stim-Orai in bleb cycling is tested by AnCoA4 and SKF96365, which are both of limited value as they are not sufficiently selective/specific. Again, what is the statistics on these pharmacological experiments? What were the concentrations of inhibitors used? This is neither stated in the text nor in the legend.

Figure 3D: what Stim1 DA construct was actually used. I guess it might be a D76A mutant? Please

state this in the methods as well as in the legend or text.

4) Inhibition of Stim-Orai function by activated ezrin:

This is an interesting and attractive hypothesis, and the authors provide indeed some supporting data. Most important is the immunoprecipitation experiment shown in Figure 4D. The authors definitely need to support this finding with a decent statistics. Can the authors show to disrupt the ezrin–Orai interaction by excess of Stim1?

Does activated ezrin or a constitutively active mutant inhibit Orai-mediated SOCE in any other cell system. The proposed functional interaction between the recombinant proteins can/should be tested in a standard expression system (HE293).

Minor and general comments:

As to the GFP-FRMD4A assay, does overexpression of this probe, which will also function as a scaffold, interfere with the blebbing cycle?

Please include appropriate statistics for all experiments.

Please provide a detailed description for all Figures: e.g. Figure 1B. I guess the x-axis displays velocity in $\mu\text{m/s}$?

Please provide a more adequate introduction to the Stim-Orai/SOCE background: line 23: “Recently, store-operated Ca^{2+} entry (SOCE) was identified as a major mechanism of calcium..... “Actually, the SOCE phenomenon was recognized long before Stim-Orai was identified as a channel complex more than a decade ago.

Reviewer#1

Summary

The authors present work that approaches a very significant area of investigation—what controls the formation and retraction of blebbing of membranes on cells. This has implications in the formation of lamellipodia and invadopodia protrusions from cells that mediate cell migration and the invasiveness of cancer cells into the cellular matrix during metastasis. The work examines the premise that calcium entry signals mediated by STIM and Orai mediates this cellular control, which would be very interesting. Unfortunately, the paper is superficial, with very preliminary and inconclusive data, and written and assembled in a way that makes little sense. The experiments are poorly designed and executed and the logic behind some of the reasoning and conclusions of the paper are very sloppy. This is a shame, since the premise of the paper seemed so significant and the authors seem to have published some good papers recently. There really are too many problems with this manuscript even to describe, so the following comments just list some of the more obvious concerns:

Comment 1

There was no detail on the statistical analysis and quantitation of the live cell imaging experiments (Fig. 1B, 1C, 1F, 1H, 2C, 2E, 2G-2J, 3E-3H, etc). It was not clear how many blebs per cell and how many cells were analyzed, or how many time the experiments were repeated. Was the difference meaningful? Are these data average of multiple cells/blebs or representative data from a single cell/bleb? Without these critical details and appropriate statistical analysis, it is impossible to determine the robustness of the findings and conclusions in the manuscript.

Response 1

In the revised manuscript, quantitative and statistical analyses were performed on all live cell imaging data. The results shown are representative of multiple experiments. The numbers of repeated experiments are specifically noted in the figure legend.

Comment 2

It was indicated that screening was performed to search for proteins specifically enriched in the expanding blebs, (page 2 bottom three lines). However, no details were provided anywhere in the manuscript regarding how the screening was performed. What was the nature and composition of the library that they used for the screening? Is this a whole genome library or a more focused library for only actin binding proteins?

Response 2

Screening was performed using an expression vector library focused on actin-related molecules as described in the previous paper (Aoki et al. PNAS 2016). We added the following sentences in the revised manuscript.

Next, we examined whether the cytoplasmic region of expanding blebs is qualitatively different from non-bleb cytoplasm in terms of protein composition. Using a library of expression vectors focused on genes involved in the regulation of actin filaments, we searched for proteins that accumulate in the bleb cytoplasm only during the expansion phase.

Comment 3

3. The logic that A23187 the experiment indicates that restricted up-regulation of calcium concentration in the cytoplasm triggers gel-sol transition of cytoplasm required for bleb expansion, is very difficult, if not impossible to follow. The gel-sol transition was never even measured! And A23187 has not been used in calcium experiments for years since it is fluorescent.

Response 3

First, as suggested by Reviewer 1, all experiments that used A-23187 were performed again with 4-Bromo-A23187 instead. In addition, we performed tracking analysis of GFP-FRMD4A on cells treated with 4-Bromo-A23187 and found that treatment with the Ca ionophore significantly increased cytoplasmic fluidity. These data were added to the revised paper as Figs. 2G-2N.

Comment 4

4. The co-IP experiments with overexpressed STIM1 and overexpressed Ezrin are somewhat amateurish and not very convincing. The same with the imaging of ER-PM junctions. At the very minimum a FRET study should have been undertaken and carefully analyzed/compared between expanding and retracting blebs.

Response 4

According to Reviewer 1's comment, we demonstrated that STIM and Ezrin competed for binding to ORAI in an expression-dependent manner by a modified co-IP experiment. The representative data and quantifications were added to the revised paper as Figs.5E and 5F.

We attempted FRET experiments but found that technical difficulties associated with the very rapid expansion of the plasma membrane could not be overcome. The velocity of membrane deformation meant that images from the two channels did not spatially match, which precluded ratiometric analysis. Instead, we performed a proximity ligation assay (PLA) on fixed cells to confirm that STIM1 and ORAI were in close proximity during bleb expansion but not during bleb retraction. These data were added as Figs. 5G and 5H in the revised manuscript.

Comment 5

5. Labels for x- and y- axis are missing in many panels.

Response 5

We added labels for x and y axis in all panels.

Reviewer#2

Summary

This paper by Aoki et al examines the role of Calcium transients in gelation-solation of the cytoplasm. They identify a pathway responsible for elevating calcium in the bleb during growth to solate the cytoplasm and one during retraction to reduce calcium to gelate the cytoplasm. The authors have generated some very nice work about the molecular control of nucleation and contractility in the past and this follows on from it. The topic is interesting and gelation/solation hasn't been examined seriously for a long time, so this is a welcome fresh new look that would clearly fit in Nature Communications. As it stands, I have several serious conceptual and technical issues with the paper which make me recommend major revisions at least. Some of these issues can be dealt with by rewriting the paper with a slightly different angle, others will need quantification and experiments.

Comment 1

-The authors state in the first line of the abstract: "the cytoplasm in mammalian cells is thought to be homogeneous". I disagree with that statement. I think it is generally accepted that the cytoplasm has both a liquid phase (the cytosol) and a solid phase. When the cell contracts, this increases pressure in the fluid phase and this can lead to detachment of the membrane from the cortex. So, although a calcium influx can lead to increased contraction, it doesn't necessarily have to solate the cytoplasm, it can just squeeze the cytosol out of the cell body to form a bleb somewhere.

Response 1

As Reviewer 2 notes, elevated Ca^{2+} ions in the cytoplasm can lead to increased intracellular pressure through activation of myosin. However, in that case, it is necessary to increase the Ca^{2+} ion in the area where acto-myosin cortex is intact other than the bleb. However, what we observed is localized elevations of Ca^{2+} ions in the cytoplasm of the expanding bleb, in which the acto-myosin cortex is absent. Thus, we concluded that the increase of calcium ions in the cytoplasm of expanding bleb plays a role in up-regulating cytoplasmic fluidity by severing actin filaments.

Comment 2

-I thinking the concept of STIM-Ora1 being involved in blebbing is interesting. What I think is much more difficult is to show that calcium mediated solation causes blebbing and decrease in calcium causes cortex assembly and gelation of the cytoplasm. Most of the work in the paper indicates a role for STIM-Ora1 in regulating calcium in blebs but how solation/gelation participates

is not well-substantiated. For example, when a bleb forms, the membrane detaches from the cortex and the cortex remains at the base of the growing bleb for a few seconds (see Zatulovskiy E et al, JCB, 2014 for example). Because the cortex remains for a few seconds (in fact most of the growth), an alternative explanation is that large macromolecules/organelles are excluded from the bleb because of sieving by the cortex. After a while, the cortex disassembles, probably because actin nucleators are membrane linked, and the barrier preventing macromolecules/organelles from entering into the bleb is removed. Once the barrier is removed, a change in cytoplasmic properties will take place because organelles, macromolecules etc can penetrate into the bleb. In fact, this phenomenon can also be observed by looking at the refractive index of blebs: when they grow they are very transparent and phase clear; while when they retract they are darker. ER probably penetrates into the blebs because it is often found very closely associated to the cortex/membrane and because its thin tube-like structure allows it to deform and get dragged into the bleb by the flow of cytosol. Currently, this sieving hypothesis is not at all considered and not shown incorrect.

Response 2

First, we performed tracking analysis of GFP-FRMD4A on cells treated with 4-Bromo-A23187 and found that treatment with the Ca ionophore significantly increased cytoplasmic fluidity. These data were added to the revised paper as Figs.2H-2K.

As Reviewer 2 says, the actin cortex appears to remain at the base of the growing bleb during the early stages of bleb formation. However, the remnant actin cortex was observed only at certain focal planes by scanning confocal microscopy, and it was not at all clear whether a complete actin network actually exists at the base of the bleb. Therefore, we used a spinning disc confocal microscope to reconstruct the 3D structures of the actin cortex and the endoplasmic reticulum at the earliest stages of bleb expansion. As shown in Fig.S4 in the revise manuscript, we observed substantial gaps in the remnant actin cortex through which the endoplasmic reticulum freely flowed into the cytoplasm of expanding bleb. Therefore, we concluded that the remnant actin cortex does not maintain a fine and complete actin meshwork that could function as a sieve.

Comment 3

-In general, my reading of the paper is that: 1) an initial calcium influx may lead to the initial contraction, 2) continued influx means that the bleb appears to have more calcium, 3) excess calcium is cleared by STIM-Ora1. I think that showing the gelation/solation part convincingly would require a lot more work and careful consideration of the sieving hypothesis. In general, I find the experiments to support the sol-gel hypothesis unspecific: latrunculin treatment is a very blunt tool.

Response 3

The influx of Ca^{2+} ion is not limited to bleb initiation but rather persists throughout expansion. Since the expanding bleb lacks the acto-myosin cortex and Ca^{2+} ion is up-regulated only in the cytoplasm of expanding bleb, contractile force cannot be actuated during this time to increase intracellular pressure. Calcium ion is imported from the extracellular space in a passive manner down the

concentration-gradient by Orai1, which we show is preferentially activated by STIM1 in the expanding bleb that lacks the inhibitory actomyosin cortex. We have not seen evidence for the efflux of intracellular Ca^{2+} ion by the STIM-Orai complex, as claimed by the reviewer, in our review of the literature. While we cannot completely discount the possibility that the remnant actin cortex functions as a sieve, our evidence outlined in Response 2 does not justify consideration of the sieving hypothesis at this time.

Regarding the use of Latrunculin A, we would share the reviewer's reservation if our intention was to examine the role of the acto-myosin cortex itself in gel-sol conversion. However, we used Latrunculin A for the express purpose of experimentally creating a condition to test whether plasma membrane free of the actin cortex, such as found in expanding blebs, could promote STIM-Orai interaction. In reviewing our manuscript, we found that the statement on Page 6, line 7-8 "... resulting in impaired bleb retraction") could be misconstrued as an overly comprehensive interpretation. We have redacted the statement in the revised text.

Page 6 Line 11

Accordingly, treatment of cells with the actin polymerization inhibitor Latrunculin B increased the area of ER-PM contact sites and the cytoplasmic calcium concentration (Figs. 5B, 5C and Movie S7).

Comment 4

- The authors show that the fluorescence intensity of GCamp6 is higher in the bleb than in the rest of the cell. This could either be due to increased calcium in the bleb or it could be because the bleb interior has less excluded volume because there are no organelles or macromolecules when it grows. In movies, examining cytoplasmic GFP, it is not uncommon to get the impression there is more GFP in the blebs than in the cell body because of these effects. In fact, you can see an increase in the fluorescence intensity of RFP on S1E. To examine this carefully, the authors should normalize fluorescence intensity to that in the cell body and plot the evolution of RFP fluorescence through bleb. The authors should then compare the fluorescence increase in GCamp (similarly normalized) to the fluorescence change for RFP in the blebs (like they have done for VASP/PIP5Kgamma – except that they should normalize to the fluorescence in the cell body). This would clearly exclude the possibility of an artefact due to differences in excluded volume in the cell body vs the bleb and it would convincingly show that we are looking at increased Ca^{2+} . From the manuscript figures, it looks like they have done the experiments and they would just need to quantify this.

-I have a similar comment about retraction: because macromolecules and organelles can penetrate in the bleb once the old cortex at the base has been removed, the excluded volume will increase and the fluorescence of a cytoplasmic RFP would decrease. So again, comparing GCamp fluorescence change to mCherry fluorescence change would be good.

Response 4

According to the comment of Reviewer 2, we expressed GFP-Mena, GFP-VASP, GCaMP6 vectors together with RFP vectors for normalization and then compared the fluorescence increase in GFP or GCaMP6 to the fluorescence change for RFP in the blebs. We added Figs. 1K, 2D, S2C and S2F to demonstrate the quantitative change over time in the ratio of the fluorescence intensities during bleb cycle. We could not detect the effects of excluded volume as judged by the time course changes of RFP signals.

Comment 5

-Can the authors observe increases in GCaMP6 fluorescence in the cell body? It would be surprising that Ca²⁺ only increases in the bleb but not in the cell body. Surely there is no barrier stopping calcium from diffusing into the cell body from the bleb.

Response 5

The fluorescence intensity of GCaMP6 is increased only in the cytoplasm of expanding blebs. A similar phenomenon was reported in dendrites that have received a neurotransmitter at the synapse of a neuron. After stimulation by a neurotransmitter, elevation of Ca ions occurs within the cytoplasm of dendrites (Sabatini et al. *Neuron* 2002). However, the molecular mechanism of the barrier that prevents diffusion of Ca ions from the dendrite into the cytoplasm is still unknown. Currently, we are studying the mechanism that prevent the diffusion of Ca ions from the expanding blebs to the cell body but the content is beyond the scope of this paper.

Comment 6

-On Fig 1D, the bleb indicated by an arrow shows the phenomenon convincingly (i.e. high Mena during expansion, low Mena during retraction) but the one just above it does: it has low Mena fluorescence then this increases as it retracts.

Response 6

Live imaging of membrane blebbing in GFP-Mena-expressing cells was repeated. We regret that the original figure did not clearly reflect our assertion and have replaced it with a more representative figure in the revised manuscript (Fig. 1G).

Comment 7

-I find the effect of A23187 difficult to judge since it first makes large blebs but eventually floods the system with calcium and stops blebbing. The blebs appear homogeneously fluorescent in GCaMP6, yet retraction seems to still proceed with no problem for a while.

Response 7

We repeated live imaging of membrane blebbing of cells treated with Ca ionophore and replaced the original data with the representative one in the revised manuscript. Treatment with Ca ionophore caused a more gradual membrane bleb retraction phase, the consequence of which was larger blebs than in control cells and eventually suppression of blebs (Fig. 2G).

Comment 8

-I also find the effect of Ezrin T567E a bit difficult to judge because the cells don't bleb. Can the authors trigger some blebs with a low dose of cytochalasin?

Response 8

As per the reviewer's suggestion, we treated the Ezrin-T567E-expressing cells with a low dose of cytochalasin, which triggered the formation of small blebs. We found that the PM-ER contact sites failed to form in such blebs, suggesting that activated Ezrin inhibits STIM-Orai interaction to prevent Ca ion uptake, which suppresses bleb expansion (Figs. S6A-S6E).

Comment 9

-In Fig 1C, the authors should quantify and compare the Mean square displacement of the particles.

Response 9

We measured particle motion quantitatively and compared the mean square displacement and the diffusion coefficient of the particles. The result of this analysis was added as Figs. 1E and 1F in the revised manuscript.

Comment 10

-Fig 2H, 3F need to be quantified in some way to allow for the comparisons of the two conditions to be reliably made.

Response 10

We quantified the average retraction velocity of the plasma membrane during the retraction phase and compared the two experimental conditions by appropriate statistical methods (Figs. 2L-2N, 4D-4F, 4J-4L, S3F-S3H and S5C-S5E).

Comment 11

-Fig 2D, the CAAX and the GCamp signal appear to switch polarity in the overlays. We have green distal and red proximal in the first panel but red distal and green proximal in the last. Why do these swap position? In 2E, the authors should show the whole time course from nucleation to retraction.

Response 11

This is due to the nature of image acquisition where a certain focal plane is first scanned in one channel (red) and then in the other (green). As the plasma membrane deforms very rapidly during blebbing, there is a time lag in image acquisition between red and green, so the green signal appears more distant than the red signal during the expansion phase, and conversely, during the retraction phase, the green signal appears more proximal than the red signal.

We repeated the experiment of Fig. 2E of the original manuscript and replaced the original data with the new data to show the entire time course from nucleation to retraction (Fig. 2E).

Reviewer#3

Summary

This manuscript reports on a tight linkage between the Stim-Orai pathway of store-operated Ca^{2+} entry and the process of membrane blebbing in DLD1 cancer cells. As membrane blebbing is an essential event in amoeboid migration, the authors address a timely and important aspect of Stim-Orai pathophysiology. It is convincingly demonstrated that bleb expansion and retraction is associated with local changes in cytosolic Ca^{2+} within the blebs and involves a Ca^{2+} entry mechanism. Using standard pharmacological tools and expression of a constitutively active Stim1 construct, the authors provide evidence for involvement of Orai-mediated Ca^{2+} entry and for negative regulation of Stim-Orai interaction by activated ezrin. The authors propose that ezrin is critically involved in both reassembly of cortical actin during bleb retraction and suppression of Stim-Orai-mediated Ca^{2+} signaling by competitive interference with Stim-Orai communication. Especially this latter conclusion appears largely speculative and premature. In general, the experimental approaches used to probe Orai signaling are barely adequate. This part of the study needs convincing complementation by more refined experiments as outlined below. Moreover, most results are insufficiently documented, especially statistical information is incomplete. More comprehensive description of the experiments and illustrations is required.

Comment 1

1) Orai-mediated bleb cycling in DLD1:

The authors conclude that the proposed Orai-mediated control of bleb cycling may be a key mechanism in cancer cell migration and invasion. Although the linkage between blebbing and invasion is well established, the authors need to show that modulation of Orai function inhibits not only bleb formation but also Ca^{2+} levels within blebs and migration? For this purpose, specific genetic manipulation of Orai function such as expression of a dominant negative Orai mutant (E106Q; see also below) should be used instead of standard, barely specific inhibitors. Also the effects of constitutively active ezrin on bleb Ca^{2+} levels and migration need to be analyzed.

Response 1

According to the comment of Reviewer 3, we examined the effect of over-expression of dominant negative form of Orai1E106Q on the bleb behavior and cell migration. Expression of Orai1E106Q

mutant led to a decrease in bleb size and shortened the course of bleb expansion (Figs. 4A-4F). Furthermore, we confirmed that DLD1 cells overexpressing Orai1E106Q exhibited markedly reduced migration by the Boyden chamber migration assay (Fig. 6A). Conversely, expression of constitutively active mutant of Ezrin suppressed bleb formation and cell migration completely (Figs. 5J and 6A).

We utilized the fluorophore sensor GCaMP6 to visualize spatial and temporal changes in Ca ion concentration. As the raw fluorescence intensity of GCaMP6 varies depending on protein expression level, it is technically difficult to measure absolute Ca ion concentration or to compare the fluorescence intensity of GCaMP6 among different cells. In order to compare the Ca ion level in control DLD1 cells with those in cells expressing Orai1E106Q, we normalized the fluorescent intensity of GCaMP6 in the bleb by that in the cell body, which was invariant. The results are presented as Fig. 4C in the revised manuscript.

Comment 2

2) Orai-mediated Ca²⁺ signaling and solation in expanding blebs:

In the abstract it is stated that “the solation of the cytoplasm is caused by a sharp rise in the calcium concentration“. I don't see evidence for such causal relation. For this statement one needs to show that fluidity of the bleb cytoplasm (GFP-FRMD4A assay) is indeed affected by modulation Ca²⁺ entry into the bleb.

Response 2

This point is the same as Reviewer1's comment 3. We performed tracking analysis of GFP-FRMD4A on cells treated with Ca ionophore, and found that treatment with Ca ionophore significantly increased cytoplasmic fluidity. These data were added to the revised paper as Figs. 2G-2N.

Comment 3

Solation is certainly required for bleb expansion, but, at this stage, one may only conclude that this process is associated with increased Ca²⁺ in the bleb. Moreover, the DLD1 cells appear to display a significant level of constitutive blebbing. Accordingly, in Figure 2A, the experiment starts at time 0 with an almost maximally expanded bleb that shows Ca²⁺ levels probably way above that in the rest of the cytosol. In Figure 2A, this expansion and Ca²⁺ levels then increases moderately for about 10s. In Figure 2C, the time course of Ca²⁺-sensitive fluorescence in the bleb is apparently compared to the fluorescence of NucBlue and labeled with „cytosol“. This is misleading since one would expect a comparison with the cytosolic Ca²⁺ outside the bleb reported by GCaMP6s. How would a comparison of cytosolic Ca²⁺ within and outside the bleb look? What is the statistics for these experiments. For how many cells/blebs are the displayed images in fact representative? In general, information on experimental statistics is incomplete throughout the paper. The presented images give the impression that constitutive bleb cycling is associated with a constantly elevated

Ca²⁺ level in the bleb. Based on what is known about Stim1-Orai1 function, it is difficult to conceive that this excessive, tonic Ca²⁺ elevation is mediated by Orai as this influx typically shows Ca²⁺-mediated negative feedback regulation. It appears required to show that intra bleb Ca²⁺ is reduced by specific suppression of Orai function. This can be achieved by either overexpressing just Orai1 (to disrupt Stim-Orai stoichiometry) or by a dnOrai (E106Q).

Response 3

In Figs. 2C and 2D, fluorescent intensities of GCaMP6 in the bleb and in the cell body were normalized to the fluorescent intensities of co-expressed RFP in the respective regions. The presentation of our results was altered to avoid unnecessary confusion. As shown in Fig. 2D, Ca²⁺ ion concentration in the bleb is elevated from the early stages of bleb expansion but decreases during bleb retraction. We found that STIM-Orai interaction is abrogated concurrent with bleb retraction. It is interesting to speculate on the possibility of a Ca²⁺ ion-mediated negative feedback mechanism to regulate STIM-Orai interaction, as the reviewer points out, and we plan to address this topic in future works. Please refer to Response 1 above for the results of the experiment using dominant negative Orai1 (E106Q).

Comment 4

3) Stim-Orai is accumulated with PM-ER junctions in blebs:

Figure 3 aims to demonstrate that Stim-Orai complexes (punctae) are formed in an expanding bleb. Unfortunately, these illustrations are all but convincing. Again, what is the statistics on these experiments? The Images illustrate that the ER appear to detach from the PM in the retracting bleb. Figure 2B also shows that there is abundant colocalization and probably interaction of Stim and Orai in the basal area of the bleb, while only a few “points” of co-localization are highlighted (arrows) by the authors in the bleb membrane. More convincing would be to demonstrate interaction by FRET.

A role of Stim-Orai in bleb cycling is tested by AnCoA4 and SKF96365, which are both of limited value as they are not sufficiently selective/specific. Again, what is the statistics on these pharmacological experiments? What were the concentrations of inhibitors used? This is neither stated in the text nor in the legend.

Response 4

We tried several approaches to demonstrate the co-localization of STIM1 and Orai1 only at the expanding bleb more convincingly. As we detailed in response to Reviewer 1's comment 4, FRET experiments were technically difficult due to the very rapid expansion of the plasma membrane. Because the membrane deformed quickly, the two color images did not match, and ratiometric analysis could not be performed. The results of the proximity ligation assay (PLA) performed instead are added as Figs. 5G and 5H in the revised manuscript. We repeated the experiment using pharmacological inhibitors (N=20) and performed statistical analysis in the revised manuscript (Figs. 3D-3F). We added the statement about the details of pharmacological experiments as follows.

Page 14 Line 8

(D) The Average number of membrane blebs in DLD1 control cells and 50 μ M AnCoA4 or 10 μ M SKF96365-treated cells during 10 min were quantified. The SD was calculated based on the values from N=20 independent cells. ****P < 0.0001 (One-way ANOVA).

(E) The area of membrane blebs in DLD1 control cells and 50 μ M AnCoA4 or 10 μ M SKF96365-treated cells during 10 min were quantified. The SD was calculated based on the values from N=20 independent blebs. ****P < 0.0001 (One-way ANOVA).

(F) The retraction velocities of membrane blebs in DLD1 control cells and 50 μ M AnCoA4 or 10 μ M SKF96365-treated cells were quantified. The SD was calculated based on the values from N=20 independent blebs. ****P < 0.0001, ***P < 0.001 (One-way ANOVA).

Comment 5

Figure 3D: what Stim1 DA construct was actually used. I guess it might be a D76A mutant? Please state this in the methods as well as in the legend or text.

Response 5

We added the statement about the characteristics of dominant-active form of STIM1 as follows.

Page 5 Line 19

By contrast, the expansion phase was prolonged, and bleb size increased, when the constitutive active mutant of STIM-1 (STIM1 D76A) was expressed (Figs.4G-4L).

Comment 6

4) Inhibition of Stim-Orai function by activated ezrin:

This is an interesting and attractive hypothesis, and the authors provide indeed some supporting data. Most important is the immunoprecipitation experiment shown in Figure 4D. The authors definitely need to support this finding with a decent statistics. Can the authors show to disrupt the ezrin–Orai interaction by excess of Stim1?

Response 6

According to Reveiwer3's comment, we demonstrated that STIM and Ezrin competed for binding to ORAI in an expression-dependent manner by improved co-IP experiment. These data were added to the revised paper as Figs. 5E and 5F.

Comment 7

Does activated ezrin or a constitutively active mutant inhibit Orai-mediated SOCE in any other cell system. The proposed functional interaction between the recombinant proteins can/should be tested in a standard expression system (HE293).

Response 7

We found that cells expressing Scarlet-EzrinT567E were insensitive to Thapsigargin treatment, such that they did not show elevated Ca²⁺ ion levels as in control cells (Figs. S6F and S6G). We also tested the regulation of Orai-mediated SOCE by ezrin in the epithelial cell line MDCKII (Figs. S6H and S6I). We are working to elucidate the molecular mechanism SOCE regulation by ezrin as we agree that it is an important topic of study. However, as this paper is focused on the roles of SOCE on membrane blebbing, we believe the detailed analysis of Ezrin's role in the regulation of SOCE in other experimental conditions is beyond the scope of this study.

Comment 8

As to the GFP-FRMD4A assay, does overexpression of this probe, which will also function as a scaffold, interfere with the blebbing cycle?

Response 8

We confirmed that expression of GFP-FRMD4A itself did not affect the dynamics and behavior of bleb (Fig. S1).

Comment 9

Please include appropriate statistics for all experiments.

Response 9

Thorough statistical analysis was performed for all experiments.

Comment 10

Please provide a detailed description for all Figures: e.g. Figure 1B. I guess the x-axis displays velocity in $\mu\text{m/s}$?

Response 10

We added labels for x and y axis in all panels.

Comment 11

Please provide a more adequate introduction to the Stim-Orai/SOCE background: line 23: “Recently, store-operated Ca²⁺ entry (SOCE) was identified as a major mechanism of calcium..... “Actually, the SOCE phenomenon was recognized long before Stim-Orai was identified as a channel complex more than a decade ago.

Response 11

We removed ‘Recently’ from this introductory sentence about SOCE.

Reviewer #1 (Remarks to the Author):

Overall, the paper is improved particularly with respect to the comments raised by the other reviewers. The authors added the important experiments to examine the action of the Orai dominant negative, E106Q. Although an important experiment and the results of expression are clear, it's not clear that they did the control transfection, just seems they used WT cells. That's not good. They should also have done the WT-Orai1 expression as a control. That also would have important dominant negative effects.

In response to my criticisms, the authors have tried, but not really succeeded in adding more convincing data. They did improve the statistical analysis, which was important. But they fell short on being able to do FRET analysis of the STIM-Orai interaction in their cells. This is a pity, but the authors claimed they were unable to do this due to the speed of the effect which did not quite make sense. Instead they attempted to do a proximity ligand assay which could have been good. However, the description of these results was difficult for me to understand at all. The description of the experiments in Fig. 5 in both the text and in the figure legend, did not allow me to make sense of these experiments. There were few details given on what they had actually done and what the images showed or meant.

Reviewer #2 (Remarks to the Author):

In this revision, the authors have made a significant effort to quantify their data and have added a number of experiments to address my comments and those of other reviewers. However, there still remain substantial issues with the paper which prevent me from recommending publication.

Major issues:

-I still find the idea that bleb emergence is due to solation and retraction due to gelation unsubstantiated. If blebs were due to solation, then we should observe an increase in the diffusion of tracers in the cytoplasm below where a bleb is about to emerge. Once the bleb has emerged, it is expected that the contents of the bleb are less dense than the cell body because of sieving. The images of penetration of some ER into blebs are convincing but the vast majority of the ER stays in the cell body until the actin cortex has disassembled (see Fig 3A, most of the ER is kept out until about 35s when it finally penetrates into the bleb because the cortex at the bleb base has disappeared). Furthermore, to be really convincing in their measurements of particle movement, the authors should take into account convective flows. Indeed, bleb growth is faster than bleb retraction. So it is possible that the faster movement of particles in growing blebs is just due to convective flows. Furthermore, if the aggregates are going to be used as a probe to measure diffusion constants, they must have a fairly narrow size distribution. Overall, I feel gelation-solation is a distraction from the data about Stim-Orai1 and because the gelation-solation part is poorly substantiated, it undermines the rest of the paper.

-If calcium were the cause of solation, you would expect high calcium in a localised region of the cell body just before a bleb starts growing.

-In writing, the authors need to clearly define the cytoplasm which has a fluid phase and a solid phase. I think cytoplasm sol and cytoplasm gel are misleading. The gel part is not strictly just a

cytoskeletal gel, it also comprises membranes (like the ER) and macromolecules like ribosomes or mitochondria.

Minor issues:

-Some of the units on graphs need to be changed: 1F: the diffusion coefficient should be in $\mu\text{m}^2/\text{s}$; 2F: I don't really understand how you can compute a ratio of fluorescence membrane compared to cytosol for a membrane marker; 2L, 3D, 4D, 4J: give the number of blebs: is this per cell? Per unit area? Per unit time?

Reviewer #3 (Remarks to the Author):

The authors did a good job in amending the manuscript. Most issues were adequately addressed by either experiment or discussion. Nonetheless, I have still some questions/requests and a minor comment:

1) The issue of STIM-Orai signaling being exclusively activated in expanding blebs is indeed intriguing but conceptually still puzzling. The authors demonstrate a fairly exclusive STIM-Orai interaction in expanding blebs by proximity ligation. This is a crucial experiment. Can the authors provide any reasonable negative and/or positive control for this assay/cell system? Do the authors assume that ER depletion, as a basis of STIM activation, is prominent in expanding blebs or alternatively that the ER filling state is similar throughout the cells. Would thapsigargin-induced store depletion trigger PLA signals in other parts of the cell? It is important to provide at least some answers to these questions, preferentially by experiments but at least by a plausible discussion.

2) It may not be necessary to have a "Results and Discussion" section and a separate Discussion in addition.

Reviewer#1

Summary

Overall, the paper is improved particularly with respect to the comments raised by the other reviewers.

Comment 1-1

The authors added the important experiments to examine the action of the Orai dominant negative, E106Q. Although an important experiment and the results of expression are clear, it's not clear that they did the control transfection, just seems they used WT cells. That's not good. They should also have done the WT-Orai1 expression as a control. That also would have important dominant negative effects.

Response 1-1

According to the comment of Reviewer 1, related experiments were performed again using the wild-type expression vector of Orai 1 as a control. These results are shown in Figure 4 in the revised manuscript.

Comment 1-2

In response to my criticisms, the authors have tried, but not really succeeded in adding more convincing data. They did improve the statistical analysis, which was important. But they fell short on being able to do FRET analysis of the STIM-Orai interaction in their cells. This is a pity, but the authors claimed they were unable to do this due to the speed of the effect which did not quite make sense. Instead they attempted to do a proximity ligand assay which could have been good. However, the description of these results was difficult for me to understand at all. The description of the experiments in Fig. 5 in both the text and in the figure legend, did not allow me to make sense of these experiments. There were few details given on what they had actually done and what the images showed or meant.

Response 1-2

In FRET analysis, it is necessary to acquire images of fluorescent proteins excited at two different wavelengths at the same location in the cell and calculate the ratio of fluorescence brightness derived from those images. However, in the expanding bleb, the cell membrane deforms very rapidly, so the image acquired by the first excitation light and the image acquired by the second excitation light do not match spatially. This is evident in the snapshot shown in Figure 2E. This figure is not the result of FRET analysis, but GCaMP-CAAX and mCherry-PH (both always co-localized at plasma membrane) did not co-localize even when captured with a high-speed spinning disk microscope. In this figure, the green signal appears more distal than the red signal during the expansion phase because of the time lag in image acquisition between red and green channels. Thus, it is technically impractical to prove the co-localization of two proteins using FRET analysis

in the expanding bleb. Therefore, in this study, we used the Proximity Ligation Assay (PLA), which is often used to show that two proteins form a complex in situ in fixed cells. PLA is a commonly used experimental technique in the field of cell biology and its detailed explanation is not repeated in most papers, so we have added a reference to the original article reporting the PLA method (Fredriksson et al. *Nat Biotechnol* 2002) to the revised manuscript. In addition, a paper showing Orai1 and STIM1 in complex using the PLA assay was recently reported (Secondo et al. *Stroke* 2019).

Reviewer#2

Summary

In this revision, the authors have made a significant effort to quantify their data and have added a number of experiments to address my comments and those of other reviewers. However, there still remain substantial issues with the paper which prevent me from recommending publication.

Comment 2-1

I still find the idea that bleb emergence is due to solation and retraction due to gelation unsubstantiated. If blebs were due to solation, then we should observe an increase in the diffusion of tracers in the cytoplasm below where a bleb is about to emerge. Once the bleb has emerged, it is expected that the contents of the bleb are less dense than the cell body because of sieving. The images of penetration of some ER into blebs are convincing but the vast majority of the ER stays in the cell body until the actin cortex has disassembled (see Fig 3A, most of the ER is kept out until about 35s when it finally penetrates into the bleb because the cortex at the bleb base has disappeared).

Response 2-1

We do not claim that cytoplasmic solation induces bleb formation. In order to dispel any ambiguity, we added a new schematic of the molecular mechanism of bleb expansion as Figure 4M in the revised manuscript. First, blebs are formed by the protrusion of the PM according to intracellular pressure at sites of local defect in the actin cortex. As shown in Supplementary Figure 4, a patchy defect in the actin cortex is observed during the earliest stages of bleb formation, at which point the ER-PM contact sites are formed. The ER is then propelled into the bleb cytoplasm during the subsequent expansion phase, throughout which the ER-PM contact sites are maintained. The formation of ER-PM contact sites is responsible for the influx of calcium ion from the extracellular space via SOCE, which is required to increase the cytoplasmic fluidity of the expanding bleb since treatment of cells with SOCE inhibitors significantly impairs bleb expansion (Fig. 3C). The increase of cytoplasmic fluidity in the expanding bleb enables the rapid expansion of the plasma membrane; it is not the cause of blebbing.

Comment 2-2

Furthermore, to be really convincing in their measurements of particle movement, the authors should take into account convective flows. Indeed, bleb growth is faster than bleb retraction. So it is possible that the faster movement of particles in growing blebs is just due to convective flows. Furthermore, if the aggregates are going to be used as a probe to measure diffusion constants, they must have a fairly narrow size distribution. Overall, I feel gelation-solution is a distraction from the data about Stim-Ora1 and because the gelation-solution part is poorly substantiated, it undermines the rest of the paper.

Response 2-2

We initially used intracellular aggregates formed by exogenously expressed GFP-FRMD4A protein to measure cytoplasmic fluidity in the original version of our manuscript, but we repeated all the measurement of cytoplasmic fluidity using quantum dots again because they have a more defined size distribution. Quantum dots was used to measure cytoplasmic fluidity in a previous study (Grady et al. *Soft Matter* 2017). As Reviewer 2 points out, it is difficult to completely ignore the effects of convective flows in the measurement of cytoplasmic fluidity, but we compared the cytoplasmic fluidity at the late stage of expanding bleb and at the beginning of retracting blebs to minimize the effects of convective flows (Fig. 1C and movie S1). Here, the period after the maximum expansion of the bleb and before the start of reassembly of actin cortex is defined as the late stage of expansion phase; the period immediately after reassembly of the actin cortex is defined as the early stage of retraction. As shown in Figs. 1C-1F, the movement of quantum dots decreases sharply after the reassembly of the actin cortex in the retracting blebs.

Comment 2-3

If calcium were the cause of solution, you would expect high calcium in a localised region of the cell body just before a bleb starts growing.

Response 2-3

This comment is related to Comment 3-1 of Reviewer 3. The luminal calcium levels of ER closely associated with the plasma membrane decreases just before bleb expansion. Please see Comment 3-1 for more explanation.

Comment 2-4

In writing, the authors need to clearly define the cytoplasm which has a fluid phase and a solid phase. I think cytoplasm sol and cytoplasm gel are misleading. The gel part is not strictly just a cytoskeletal gel, it also comprises membranes (like the ER) and macromolecules like ribosomes or mitochondria.

Response 2-4

We agree with Reviewer 2's comments. The expressions such as “cytoplasm sol” and “cytoplasm gel” are ambiguous and confusing, so we removed these expressions throughout the manuscript and replaced them with descriptors of the changes in cytoplasm fluidity, i.e. increase/decrease.

Comment 2-5

Some of the units on graphs need to be changed: 1F: the diffusion coefficient should be in $\mu\text{m}^2/\text{s}$; 2F: I don't really understand how you can compute a ratio of fluorescence membrane compared to cytosol for a membrane marker; 2L, 3D, 4D, 4J: give the number of blebs: is this per cell? Per unit area? Per unit time?

Response 2-5

We thank Reviewer 2 for pointing out the inadequate labeling of our graphs. All corrections have been made following the suggestions of Reviewer 2. We corrected the unit of the diffusion coefficient to $\mu\text{m}^2/\text{msec}$ (Fig. 1F).

Regarding Fig. 2F, as Reviewer 2 correctly noted, the description of the image analysis was incorrect. The ratio of fluorescence intensities of GCaMP-CAAX and mCherry-PLC δ -PH at the cell membrane was quantified. We corrected the figure legend of Fig.2F as follows.

Page 13 Line 17

(F) Quantification of the experiments shown in (E). Fluorescence intensities of GCaMP6s-CAAX and mCherry-PLC δ -PH were measured at the bleb cell cortex (membrane) and their ratio (GCaMP6s-CAAX / mCherry-PLC δ -PH) were plotted against time. The SD was calculated based on the values from N = 5 independent experiments.

Regarding Figs. 2L, 3D, 4D and 4J, we counted the number of blebs formed over 10 minutes per cell. We added this information in each figure legends as follows.

Page 14 Line 6

(L) The average number of membrane blebs in DLD1 control cells and 4-bromo-A23187-treated cells **over 10 min per cell** were quantified. N=20. The SD was calculated based on the values from three independent experiments. ****P < 0.0001 (Student's t test).

Reviewer#3

Summary

The authors did a good job in amending the manuscript. Most issues were adequately addressed by either experiment or discussion. Nonetheless, I have still some questions/requests and a minor comment:

Comment 3-1

The issue of STIM-Orai signaling being exclusively activated in expanding blebs is indeed intriguing but conceptually still puzzling. The authors demonstrate a fairly exclusive STIM-Orai interaction in expanding blebs by proximity ligation. This is a crucial experiment. Can the authors provide any reasonable negative and/or positive control for this assay/cell system? Do the authors assume that ER depletion, as a basis of STIM activation, is prominent in expanding blebs or alternatively that the ER filling state is similar throughout the cells. Would thapsigargin-induced store depletion trigger PLA signals in other parts of the cell? It is important to provide at least some answers to these questions, preferentially by experiments but at least by a plausible discussion.

Response 3-1

First, we added positive control (STIMATE) and negative control (Claudin) data for the PLA assay in Figure 5. STIMATE, a component of the store-operated calcium entry complex, is closely associated with STIM1 (Jin et al. *Nat Cell Biol.* 2015). The PLA signal between STIM and STIMATE was observed throughout the bleb cycle (Fig. S5A). By contrast, there was no appreciable PLA signal between STIM1 and claudin-3, as a membrane protein other than Orai1, at the plasma membrane (Fig. S5A). This indicates that STIM1 does not interact with the plasma membrane at sites other than the ER-PM contact site.

Second, we visualized changes in ER luminal calcium levels during the bleb cycle using G-CEPIAer, a green fluorescent indicator for calcium imaging in the ER (Suzuki et al. *Nat Commun* 2014) (Figure R1; please see next page). Just before the initiation of bleb expansion, the ER luminal calcium level actually decreased. Subsequently, the ER luminal calcium levels gradually increased over the course of the expansion phase. This result suggests that the transient depletion of calcium ion in the ER triggers the ER-PM contact site formation through the activation of STIM1. As shown in Figure R1-A, the luminal calcium levels within the cell was not uniform. The ER luminal calcium levels in the expanding blebs are relatively high (black arrowheads), but the luminal calcium levels in the ER near the plasma membrane just before bleb formation is comparatively suppressed (white arrowheads). This heterogeneous distribution of ER luminal calcium ion is very interesting, but a detailed analysis is clearly beyond the scope of this paper; we would like to clarify it in a future work.

Finally, we examined the effects of thapsigargin treatment on the ER-PM contact site formation. As shown in Supplemental Figures 5B and 5C, PLA signals between STIM1 and Orai1 increased in cells treated with thapsigargin. This observation is in good agreement with our result that thapsigargin treatment induces an increase in calcium ion in the cytoplasm and enlarges bleb size (Fig. 2O).

Figure R1 Aoki et al.

The change in ER luminal Ca^{2+} concentration during membrane blebbing

(A) A pseudocolor image of the fluorescent intensity of G-CEPIA1er in DLD1 cell. White arrowheads show low Ca^{2+} concentration ER region before bleb formation, and black arrowheads show high Ca^{2+} concentration ER region in retracting blebs. Result shown is representative of three independent experiments. (Scale bar: 10 μm .)

(B) A pseudocolor image of the fluorescent intensity of G-CEPIA1er during membrane blebbing. Result shown is representative of three independent experiments. (Scale bar: 10 μm .)

(C) Quantification of the change of fluorescence intensity of G-CEPIA1er during bleb cycle. G-CEPIA1er and mCherry-Sec61β (ER marker) were co-transfected in DLD1 cell. The fluorescence intensity of G-CEPIA1er and mCherry-Sec61β were quantitatively measured at ER in cytosol of blebs and the ratio of G-CEPIA1er to mCherry-Sec61β were plotted. The graph area colored in yellow shows the time span of the bleb lifecycle. The relative time is shown along the horizontal axis. The SD was calculated based on the values from N=5 independent experiments.

Comment 3-2

It may not be necessary to have a “Results and Discussion” section and a separate Discussion in addition.

Response 3-2

We corrected “Results and Discussion” to “Results” in the main text of the revised manuscript.

Reviewer #1 (Remarks to the Author):

No comments to authors

Reviewer #2 (Remarks to the Author):

The authors have added a number of clarifications to their manuscript which has improved it. I still have a number of issues which must be addressed before publication.

Major issues:

-The QD experiments are a nice addition to the study. The authors analyse the movement of QDs by comparing the distance of their movement and their MSD. On Fig 1D, the authors plot the movement of the QDs. They need to state: 1) the duration over which this distance is measured – this is not in the legend, 2) if they plot end-to-end distance or the sum of the frame-to-frame distance. In Fig 1E, they plot the MSDs and fit these to a linear function to determine the diffusion coefficient. On closer examination of the fits, we can see that the fit line does not always go through the origin. This is not correct and all of the fits should be redone with linear functions that pass through the origin. I presume that this will not fundamentally change the authors' conclusions but at the very least it will be correct physically. The same issue can also be found on Fig 2J. On Fig 1F, the unit of the diffusion coefficient needs to be given. This will allow readers to compare this to the diffusion coefficient expected from the Stokes Einstein relationship. The same issue is also apparent on Fig 2K.

-Page 4, calcium ionophores experiments: In this paragraph, the authors describe experiments which suddenly increase the influx of calcium leading to increased bleb size and prolonged expansion. They conclude that increase of calcium concentration increases cytoplasmic fluidity required for bleb expansion. An alternative scenario is that increased intracellular calcium increases myosin contractility (for example MLCK is calcium sensitive), which increases intracellular pressure leading to larger blebs. It may be difficult to design experiments to carefully distinguish these two hypotheses. So the authors should at least acknowledge the alternative explanation – perhaps in the discussion?

-The experiments with the ionophore show that you need intracellular calcium. The authors then examine the necessity of SOCE on page 5-6 and conclude that it is needed for fluidisation. I have a similar issue here. Blocking SOCE would decrease intracellular calcium and intracellular calcium may be needed to regulate contractility. So SOCE inhibition would decrease contractility and decrease bleb size. On page 7, the authors examine the effect of SOCE inhibition on blebs triggered by cytochalasin in cells that over-express Ezrin T567E. Here again, it is possible that intracellular calcium is necessary to regulate myosin contractility rather than cytoplasm fluidity. In this hypothesis, when SOCE is inhibited, cortical tension would be decreased and no blebs would emerge. One way of verifying this would be to measure cortical tension +/- SOCE inhibitors in the Ezrin T567E expressing cells using micropipette aspiration or AFM.

Minor issues:

-The legends for Fig 1D, 2I need to state how many blebs from how many cells were examined.

-On Fig 4I, S2B and S2E, the y-axis legend should be "number of frames".

-On Fig 5H and in its associated legend, the authors need to state what the positive and negative

controls are.

-In the discussion, the authors state that cytoplasmic fluidity must be increased synchronously with bleb growth. I do not think this is necessary, sieving alone would ensure that only the most fluid part of the cytoplasm can penetrate the blebs.

Reviewer #3 (Remarks to the Author):

With this revision, the authors indeed provide further, compelling evidence including requested controls as well as a highly plausible discussion/reasoning for their conclusions.

Reviewer #1

This reviewer has no further critiques.

Reviewer #2

Summary

The authors have added a number of clarifications to their manuscript which has improved it. I still have a number of issues which must be addressed before publication.

Response

We thank the reviewer for the careful evaluation of our manuscript and thoughtful comments.

Comment 2-1

The QD experiments are a nice addition to the study. The authors analyse the movement of QDs by comparing the distance of their movement and their MSD. On Fig 1D, the authors plot the movement of the QDs. They need to state: 1) the duration over which this distance is measured – this is not in the legend, 2) if they plot end-to-end distance or the sum of the frame-to-frame distance. In Fig 1E, they plot the MSDs and fit these to a linear function to determine the diffusion coefficient. On closer examination of the fits, we can see that the fit line does not always go through the origin. This is not correct and all of the fits should be redone with linear functions that pass through the origin. I presume that this will not fundamentally change the authors' conclusions but at the very least it will be correct physically. The same issue can also be found on Fig 2J. On Fig 1F, the unit of the diffusion coefficient needs to be given. This will allow readers to compare this to the diffusion coefficient expected from the Stokes Einstein relationship. The same issue is also apparent on Fig 2K.

Response 2-1

According to the comment of reviewer 2, we added the following information in the text. The movement of the QDs was measured over 600 msec and the sum of the frame-to-frame distance of the trajectories were plotted. The figure legends of Figs. 1D and 2H are clarified in the revised manuscript as follows:

Page 12 Line 2

Centered trajectory maps of individual QDs within the cytoplasm of expanding and retracting blebs over 600 msec (30 frames at 50 Hz). The sum of the frame-to-frame distance were plotted.

Page 14 Line 19

Individual QDs trajectories over 600 msec (30 frames at 50 Hz) within the cytoplasm of control cells and that of cells treated with 10 μ M 4-bromo-A23187 for 5 min prior to imaging. The sum of the frame-to-frame distance were plotted.

As the reviewer correctly points out, the fitted lines of the MSD plots should go through the origin. We reevaluated the data for the fit line to pass through the origin and recalculated the diffusion coefficients (D) shown in Figs. 1F and 2K as a result. The unit of the diffusion coefficient is $\mu\text{m}^2/\text{msec}$, which is now made explicit in the y-axis legends of Figs. 1F and 2K.

Comment 2-2

Page 4, calcium ionophores experiments: In this paragraph, the authors describe experiments which suddenly increase the influx of calcium leading to increased bleb size and prolonged expansion. They conclude that increase of calcium concentration increases cytoplasmic fluidity required for bleb expansion. An alternative scenario is that increased intracellular calcium increases myosin contractility (for example MLCK is calcium sensitive), which increases intracellular pressure leading to larger blebs. It may be difficult to design experiments to carefully distinguish these two hypotheses. So the authors should at least acknowledge the alternative explanation – perhaps in the discussion?

Comment 2-3

The experiments with the ionophore show that you need intracellular calcium. The authors then examine the necessity of SOCE on page 5-6 and conclude that it is needed for fluidisation. I have a similar issue here. Blocking SOCE would decrease intracellular calcium and intracellular calcium may be needed to regulate contractility. So SOCE inhibition would decrease contractility and decrease bleb size. On page 7, the authors examine the effect of SOCE inhibition on blebs triggered by cytochalasin in cells that over-express Ezrin T567E. Here again, it is possible that intracellular calcium is necessary to regulate myosin contractility rather than cytoplasm fluidity. In this hypothesis, when SOCE is inhibited, cortical tension would be decreased and no blebs would emerge. One way of verifying this would be to measure cortical tension +/- SOCE inhibitors in the Ezrin T567E expressing cells using micropipette aspiration or AFM.

Response 2-2/2-3

Since the above two comments are related, they are discussed together below. These points were also discussed in our response to the initial round of reviews (Response 1 to Reviewer #2). In the present study, we revealed that the concentration of calcium ion was selectively increased in the expanding bleb cytoplasm. As the reviewer notes, since increased intracellular calcium can enhance myosin contractility, we are observing the combined effects of increased cytoplasmic fluidity and enhanced myosin contractility in the experiments with the calcium ionophore; in the same vein, it is possible

that the decrease in intracellular calcium concentration alters bleb dynamics by decreasing not only cytoplasmic fluidity but also myosin contractility in the experiments with the SOCE inhibitors. However, it must be emphasized that the increased calcium concentration is observed only in the cytoplasm of the expanding bleb, which is devoid of the acto-myosin cortex, since we analyzed the change in bleb cytoplasm calcium concentration normalized to the calcium concentration in the cell body (Figs. 2B-2D). Therefore, contractility exerted by the acto-myosin cortex of the cell body is presumably constant throughout the time course of the bleb expansion and retraction, making it highly unlikely that this would contribute to bleb dynamics. Likewise, SOCE activation by the contact formation between the plasma membrane and the endoplasmic reticulum is also limited to the expanding bleb, which again lacks the acto-myosin cortex. From these results, we believe that it is reasonable to conclude that myosin contractility is largely unaltered at a steady state level and that its contribution to bleb dynamics is negligible.

We added the live imaging data of DLD1 cells expressing GCaMP6s and Scarlet-MRLC1 as Figure S2 in the revised manuscript to demonstrate the temporal relationship between the recruitment of myosin to the plasma membrane and changes in the cytoplasmic calcium ion concentration during bleb cycle more directly.

We have added the following statement to the text in order to make our stance on these points more explicit and to avoid any ambiguity for the readers:

Page 4 Line 8

The concentration of calcium ions in the cytoplasm was elevated during the expansion phase, but decreased during the retraction phase. It was reported that myosin regulatory light chain (MRLC) is recruited to the actin cortex during the retraction phase significantly later than other actin associated proteins⁸. The calcium ions are elevated only in the cytoplasm of expanding blebs which lacks acto-myosin cortex, and the calcium ion concentration at the time of myosin accumulation in the retracting bleb is as low as that in the cell body (Fig. S2). The calcium ion concentration in the cell body was constant throughout the time course of the bleb expansion and retraction (Figs. 2B and 2C). Therefore, we reasoned that the increase in the concentration of calcium ions in the cytoplasm of the expanding bleb is not related to the regulation of myosin contractility during the bleb cycle but to the increase of the cytoplasmic fluidity in the expanding bleb.

Page 9 Line 32

In the present study, we revealed that the concentration of calcium ion was up-regulated only in the expanding bleb cytoplasm. Since the plasma membrane of expanding blebs is devoid of the acto-myosin cortex—and since the calcium ion concentration in the cell body is unchanged at a steady state level—we propose that the increase in calcium ion during the expansion phase drives bleb dynamics by explicitly increasing the cytoplasmic fluidity in the expanding bleb.

Page 22 Line 7

Figure S2

Myosin is recruited to the plasma membrane of the bleb after the cytoplasmic concentration of calcium ions is decreased. Related to Figure 2.

Membrane blebbing of DLD1 cells transfected with GCaMP6s and Scarlet-myosin regulatory light chain 1 (MRLC1). Timing relative to the first image is indicated in white text. MRLC1 (arrowheads) was recruited to the plasma membrane during the retraction phase only after the cytoplasmic concentration of calcium ions decreased. Result shown is representative of five independent experiments. (Scale bar: 2 μ m)

Comment 2-4

The legends for Fig 1D, 2I need to state how many blebs from how many cells were examined.

Response 2-4

We revised the figure legends for Figs.1D and 2I as follows.

Page 12 Line 11

N=100 independent particles from 10 blebs from 10 independent cells.

Page 14 Line 27

N=110 independent particles from 11 blebs from 10 independent cells.

Comment 2-5

On Fig 4I, S2B and S2E, the y-axis legend should be “number of frames”.

Response 2-5

We corrected the y-axis legend for Figs. 4I, S3B and S3E.

Comment 2-6

On Fig 5H and in its associated legend, the authors need to state what the positive and negative controls are.

Response 2-6

We added the explanation of the positive and negative controls to the legend of Fig. 5H in the revised manuscript as follows:

Page 19 Line 21

The positive and negative controls are the PLA signals for the interaction between HA-tagged STIMATE (a component of the store-operated calcium entry complex, which is closely associated with STIM1) and FLAG-tagged STIM1, and between Claudin-3 (a component of cell adhesion apparatus, which is not related to SOCE) and FLAG-tagged STIM1, respectively (See also Fig. S6A).

Comment 2-7

In the discussion, the authors state that cytoplasmic fluidity must be increased synchronously with bleb growth. I do not think this is necessary, sieving alone would ensure that only the most fluid part of the cytoplasm can penetrate the blebs.

Response 2-7

As noted by the reviewer, we cannot discount sieving as a potential regulator of cytoplasmic fluidity in the expanding bleb. Therefore, the following statement was amended to the discussion in the revised manuscript:

Page 10 Line 4

As for the molecular mechanism involved in the regulation of cytoplasmic fluidity during bleb cycle, we revealed the existence of mechanisms to actively increase the fluidity of the cytoplasm by specifically up-regulating calcium ions in the cytoplasm of expanding blebs. However, the possibility remains that the severed actin filaments emanating from the disintegrating actin cortex functions as a sieve to limit particulate infusion into the expanding bleb. These possibilities need to be explored in more detail in future studies.

Reviewer #3

Summary

With this revision, the authors indeed provide further, compelling evidence including requested controls as well as a highly plausible discussion/reasoning for their conclusions.

Response

This reviewer has no further critiques.

Reviewer #2 (Remarks to the Author):

The authors have answered all of my remaining questions and addressed the issues that I raised in the previous review.

Congratulations on this interesting work!